# Linear Convergence and Generalization of FedAvg Under Constrained PL-Type Assumptions: A Single Hidden Layer Neural Network Analysis

## Abstract

In this work, we study the generalization performance of the widely adopted FedAvg algorithm for solving Federated Learning (FL) problems. FedAvg has been extensively studied from an optimization perspective under different settings; however, analyzing the generalization performance of FedAvg is particularly challenging under practical settings since it involves simultaneously bounding (1) the optimization error and (2) the Rademacher complexity of the model to be learned, which are often contradictory. Specifically, obtaining optimization guarantees for FedAvg relies on restrictive assumptions on the loss landscape, such as (strong) convexity or Polyak-Łojasiewicz (PL) inequality to be satisfied over the entire parameter space. However, for an unbounded space, it is challenging to control the Rademacher complexity, leading to worse generalization guarantees. In this work, we address this challenge by proposing novel *constrained PL-type* conditions on the objective function that ensure the existence of a global optimal to which FedAvg converges linearly after $\mathcal{O}(\log(1/\epsilon))$ rounds of communication, where $\epsilon$ is the desired optimality gap. Importantly, we demonstrate that a class of single hidden layer neural networks satisfies the proposed *constrained PL-type* conditions as long as $m > nK/d$, where $m$ is the width of the neural network, $K$ is the number of clients, $n$ is the number of samples at each client, and $d$ is the feature dimension. Finally, we bound the Rademacher complexity for this class of neural networks and establish that the generalization error of FedAvg diminishes at the rate of $\mathcal{O}(1/\sqrt{n})$.

## 1 Introduction

Federated learning (FL) is a distributed learning paradigm in which multiple clients collaborate with the help of a server to solve a joint problem while keeping the data of each client private (McMahan et al., 2017; Kairouz et al., 2021). A typical FL problem aims to solve $\min_{\mathbf{w}} \sum_{k=1}^{K} \Phi_k(\mathbf{w})$, where $\Phi_k(\mathbf{w})$ is the loss at the $k^{\text{th}}$ client and $\mathbf{w}$ refers to the joint model the clients aim to learn. A standard and widely adopted algorithm to solve the FL problem is the Federated Averaging (FedAvg) algorithm first proposed in (McMahan et al., 2017). Consequently, the study of the convergence performance of FedAvg has received wide attention (Konečný et al., 2015; Stich, 2018; McMahan et al., 2017; Li et al., 2020; Zhou & Cong, 2017b). However, when it comes to analyzing the generalization performance of FedAvg, the problem has not received significant attention, mainly due to the challenging nature of the problem (Mohri et al., 2019; Sun et al., 2023; Hu et al., 2022). To provide generalization guarantees for FedAvg, one needs to control (1) the **optimization error** (on the empirical loss) achieved by FedAvg, and (2) the **complexity measure** such as the Rademacher complexity of the FL model (Arora et al., 2019; Mohri et al., 2019; 2018). The major challenge in guaranteeing generalization performance is to upper bound both (1) and (2), which are often contradictory. In particular, proving optimization guarantees for FedAvg usually relies on restrictive assumptions on the loss landscape like (strong)-convexity (Stich, 2018; Qu et al., 2020) or Polyak-Łojasiewicz (PL) inequality to be satisfied over the entire parameter space (Haddadpour et al., 2019; Haddadpour & Mahdavi, 2019). On the other hand, the complexity measures like the Rademacher complexity grow with

the size of the parameter space Mohri et al. (2018, Theorem 5.10) leading to impractical generalization guarantees. Therefore, it is desirable to have convergence guarantees on a restricted space rather than the entire parameter space to upper bound the generalization. This leads to the question:

> **Q1:** *Can we develop conditions that are satisfied locally (on a restricted parameter space) rather than globally and provide convergence guarantees for FedAvg? Are there models that satisfy such a condition?*

The study of generalization guarantees of FL algorithms is rather limited (Mohri et al., 2019; Hu et al., 2022; Yuan et al., 2021a; Huang et al., 2024). Notably, these studies often overlook the impact of the optimization algorithms Sun et al. (2023), and often rely on assumptions like binary classification Hu et al. (2022); Mohri et al. (2019) and the Bernstein condition (Yuan et al., 2021a). Additionally, generalization bounds for meta-learning and FL are established in Fallah et al. (2021); Chen et al. (2021) under stringent assumptions such as strong convexity and bounded loss functions. Recently, Sun et al. (2023) investigated the generalization of FedAvg via the lens of uniform stability. We note that these analyses impose strong assumptions such as bounded gradients and heterogeneity on the data, which are usually not satisfied for many problems of practical interest. Moreover, the optimization guarantees provided in Sun et al. (2023) are weaker compared to the linear convergence established in our work.

Based on the above observations, we ask the following key question:

> **Q2:** *Can we provide generalization guarantees for FedAvg? If so, what is the impact of $(a)$ the number of samples per client, $(b)$ the model sizes, and $(c)$ the number of clients on the generalization performance?*

In this work, we answer the above questions affirmatively and make the following major contributions.

## 1.1 Our Contributions

**Answer to Q1.** For the first time, we establish that FedAvg converges linearly to the optimal solution (see Corollary 3.2) if the local loss functions at each client and the global loss function satisfy a novel *constrained PL-type condition* that we propose in this work (see Assumption 2.4). It is important to note that our results establish, rather than assume, the existence of at least one global optimum. To the best of our knowledge, both conditions introduced in Assumption 2.4 are novel. It is also worth noting that these conditions do not follow from any of the existing results (Chatterjee, 2022; An & Lu, 2023). In addition, we also establish that a single hidden-layer NN with squared loss satisfies the two conditions proposed in Assumption 2.4. Specifically, we establish conditions on the width of the NN as a function of the number of samples, the number of clients, the feature dimension, and the eigenvalues of the Jacobian of the loss functions (or the scaling factor of the final output layer) such that the proposed conditions are satisfied. To our knowledge, these results are novel (see Theorem 4.3).

**Answer to Q2.** To address Q2, we derive an upper bound on the Rademacher complexity for a class of single hidden layer NNs by utilizing the result that the FedAvg(GD) iterates stay within a $\rho$-ball around the initialization. We point out that this is made possible by the conditions provided in the Assumption 2.4. In particular, we show that the Rademacher complexity approaches zero if the radius $\rho = \mathcal{O}(\sqrt{n})^1$ and $m = \mathcal{O}(n^3)$, where $n$ is the number of samples at each client and $m$ is the width of the NN. We show that the generalization error regardless of the data heterogeneity diminishes as $\mathcal{O}(1/\sqrt{n})$. We finally corroborate our theoretical findings through numerical experiments.

It is worth mentioning that to address both Q1 and Q2, we *do not* make some standard assumptions that are typically used in many existing works Li et al. (2019); Stich (2018); Yu et al. (2019); Haddadpour et al. (2019); Qu et al. (2020); Woodworth et al. (2020a;b); Hu et al. (2022); Mohri et al. (2019) such as: $(i)$ (strongly) convex loss, $(ii)$ bounded loss, $(iii)$ bounded gradients, $(iv)$ bounded heterogeneity and $(v)$ existence of global optimal.

---

[1] The radius over which our new condition should be satisfied.

## 1.2 Related Work

**Convergence of FedAvg:** McMahan et al. (2017) first introduced FL to learn a global model from distributed data without sharing it with a central server. Majority of the research on FL focuses on communication-efficiency Konečnỳ et al. (2016); McMahan et al. (2017); Li et al. (2020); Smith et al. (2017) and data-privacy Bagdasaryan et al. (2020); Bonawitz et al. (2016); Geyer et al. (2017). The FedAvg algorithm serves as the foundation of most FL methods but often suffers from slow or unstable convergence under data heterogeneity Li et al. (2020); Wang & Joshi (2021). To address this, several variants have been proposed: FedProx introduces a proximal term to stabilize local updates Li et al. (2020); SCAFFOLD leverages variance-reduction to correct client drift Karimireddy et al. (2020); FedNova normalizes local updates to mitigate solution bias Wang et al. (2020); and FedPD Zhang et al. (2020) and FedLin Mitra et al. (2021) adopt primal–dual formulations to jointly address data and system heterogeneity. From a theoretical viewpoint, FedAvg has been analyzed under convex, strongly convex, and non-convex regimes, including overparameterized networks Stich (2018); Wang & Joshi (2021); Khaled et al. (2019); Yu et al. (2019). Notably, Stich (2018) proved that LocalSGD (equivalent to FedAvg) achieves linear speedup in strongly convex settings, while Zhou & Cong (2017a); Wang & Joshi (2021) extended convergence results to non-convex problems. However, most analyses rely on strong assumptions such as the PL condition over the entire parameter space Qu et al. (2021); Li et al. (2022); Song et al. (2023), which limits their applicability to a broader class of non-convex loss functions. In contrast, our analysis establishes linear convergence of FedAvg without assuming strong convexity, instead relying on a weaker condition on the loss function. This allows our results to hold for a much wider class of non-convex problems.

**Generalization of FedAvg:** The concept of generalization in centralized learning has been extensively investigated. Classical approaches rely on uniform convergence techniques based on complexity measures such as the VC dimension or Rademacher complexity to bound the generalization gap Shalev-Shwartz et al. (2010); Yin et al. (2018); Attias et al. (2018). In the context of FL, Mohri et al. (2019) derived a uniform convergence bound of order $\mathcal{O}(1/\sqrt{N})$ for agnostic FL with bounded binary loss functions, where $N$ denotes the total number of aggregated samples. Subsequent works Yuan et al. (2021b); Sun et al. (2023) refined these analyses by considering partial client participation and meta-distribution sampling. Under additional smoothness and Bernstein conditions, faster rates have been established Hu et al. (2023), but only under strong convexity and bounded loss, which limit their applicability in realistic federated systems. Sun et al. (2023) has proved a heterogeneity-dependent upper bound on the true risk of the model obtained by FedAvg, FedProx, and SCAFFOLD under the assumption of bounded gradient. To prove a better generalization of the model, we must achieve a low optimization error, preferably $\mathcal{O}(1/\sqrt{n})$ or better, which is missing in Sun et al. (2023). In this paper, we establish that the optimization error of the model trained via FedAvg converges linearly to zero. Furthermore, we show that the Rademacher complexity of the learned hypothesis class diminishes to zero as the sample size increases sufficiently large, indicating improved generalization with more data, even without assuming convexity of the loss function.

## 2 FedAvg(GD): Algorithm and Assumptions

FL aims to solve the following optimization problem:

$$\min_{\mathbf{w}} \left\{ \mathcal{L}(\mathbf{w}) \coloneqq \frac{1}{K} \sum_{k=1}^{K} \mathcal{L}_k(\mathbf{w}) \right\}, \tag{1}$$

where $\mathcal{L}_k(\mathbf{w}) \coloneqq \mathbb{E}_{(\boldsymbol{x},y)\sim\mathcal{D}_k}\ell_k(f_{\mathbf{w}}(\boldsymbol{x}),y)$ represents the expected loss at client $k \in [K]$. Here, $y \in \mathcal{Y}$ is the true label, $f_{\mathbf{w}}(\boldsymbol{x})$ is the model output parameterized by $\mathbf{w} \in \mathbb{R}^{d'}$ with $d'$ representing the dimension of parameter space, and $l_k : \mathcal{Y} \times \mathcal{Y} \to \mathbb{R}_+$ denotes the loss function used by client $k$. The data $(\mathbf{x},y)$ at the client $k$ is drawn from an unknown and potentially distinct distribution $\mathcal{D}_k$. In practical FL scenarios, the distributions $\mathcal{D}_k$ are not explicitly known. Instead each client $k \in [K]$ possesses a local dataset $\mathcal{Z}_k = \{(\mathbf{x}_{k,1}, y_{k,1}), (\mathbf{x}_{k,2}, y_{k,2}) \ldots, (\mathbf{x}_{k,n}, y_{k,n})\}$, sampled from $\mathcal{D}_k$. Therefore, we consider the empirical counterpart

---

**Algorithm 1** FedAvg(GD) McMahan et al. (2017)

1: **Initialize**: $\{\mathbf{w}_k^{0,0} = \underline{\mathbf{w}}^0\}$, $\mathbf{w}_k \in \mathbb{R}^d$ for $k \in [K]$
2: **for** $r = 0, 1, \ldots, R-1$ **do**
3:     Broadcast $\underline{\mathbf{w}}^r$ to all the clients $k \in [K]$
4:     **for** $\tau = 0, 1, \ldots, T-1$ **do**
5:         **for** each client $k \in [K]$ **do**
6:           **GD Step:** $\mathbf{w}_k^{r,\tau+1} = \mathbf{w}_k^{r,\tau} - \eta \nabla \Phi_k(\mathbf{w}_k^{r,\tau})$
7:         **end for**
8:     **end for**
9:     **Receive** $\mathbf{w}_k^{r,T}$ from nodes $k \in [K]$
10:     **Aggregation** step: $\underline{\mathbf{w}}^{r+1} = \frac{1}{K} \sum_{k \in [K]} \mathbf{w}_k^{r,T}$
11: **end for**

---

of the problem in equation 1:

$$\min_{\mathbf{w}} \left\{ \Phi(\mathbf{w}) := \frac{1}{K} \sum_{k=1}^{K} \Phi_k(\mathbf{w}) \right\}, \tag{2}$$

where $\Phi_k(\mathbf{w}) := \sum_{i=1}^{n} \Phi_{k,i}(\mathbf{w})$ and $\Phi_{k,i}(\mathbf{w}) := \ell_k(f_{\mathbf{w}}(\mathbf{x}_{k,i}), y_{k,i})$ denotes the empirical (sum) loss function at client $k \in [N]$ computed using the local samples. For notational convenience, we use the summation form of the loss to establish convergence results, although generalization bounds will later be expressed in terms of average (true and empirical) losses. A widely adopted method for solving the above optimization problem is the **Federated Averaging (FedAvg)** algorithm, introduced by (McMahan et al., 2017) (see Algorithm 1). In this and the subsequent section, we answer Q1 posed in Sec. 1. In particular, we provide a new condition in Assumption 2.4 under which the FedAvg(GD) converges to a global optimum while ensuring that the model parameters remain confined within a closed Euclidean ball of radius $\rho$. In later sections, we demonstrate that this new condition is indeed satisfied for neural networks with a single hidden layer. To establish our convergence results, we begin by introducing some standard assumptions on the (local) loss functions, formalized through the following definition.

**Definition 2.1** ($L$- Smooth). *A differentiable function $f : \mathbb{R}^d \to \mathbb{R}$ is said to be $L$-smooth if it satisfies $f(\mathbf{x}) \le f(\mathbf{y}) + \langle \nabla f(\mathbf{y}), \mathbf{x} - \mathbf{y} \rangle + \frac{L}{2} \|\mathbf{x} - \mathbf{y}\|^2$, for all $\mathbf{x}, \mathbf{y} \in \mathbb{R}^d$.*

**Assumption 2.2.** *The individual client's loss, global loss and sample-wise loss functions $\Phi_k, \Phi, \Phi_{k,i} : \mathbb{R}^{d'} \to \mathbb{R}$ for $k \in [K]$, $i \in [n]$ are $L_k, L$ and $l_{k,i}$-smooth, respectively, where $d'$ is dimension of the model. Additionally, we define the global smoothness constant as $L := \max_{k \in [K]} L_k$.*

To formalize the primary assumptions required for the convergence of FedAvg(GD) stated in Algorithm 1, we now introduce the following notion adapted from (Chatterjee, 2022).

**Definition 2.3.** *Let $f : \mathbb{R}^d \to \mathbb{R}_+$ be a continuously differentiable function defined over a closed ball $\mathbb{B}[\underline{\mathbf{w}}^0, \rho]$ centered at initialization $\underline{\mathbf{w}}^0 \in \mathbb{R}^d$ with radius $\rho > 0$. We define*

$$\alpha(\underline{\mathbf{w}}^0, \rho) := \inf_{\substack{\mathbf{w} \in \mathbb{B}[\underline{\mathbf{w}}^0, \rho] \\ f(\mathbf{w}) > 0}} \frac{\|\nabla f(\mathbf{w})\|^2}{f(\mathbf{w})}. \tag{3}$$

This quantity measures the growth of the gradient relative to the function value within the ball. Next, we state an important assumption that allows us to prove linear convergence of FedAvg(GD) within a ball around the initialization. Toward stating the assumption, we define the constant $\alpha_k(\underline{\mathbf{w}}^0, \rho)$ exactly as defined in equation 3 with $f(\cdot)$ replaced by the local loss function $\Phi_k(\cdot)$ for each client $k \in [K]$. Similarly, the constant $\alpha_g(\underline{\mathbf{w}}^0, \rho)$ is defined by replacing $f(\cdot)$ in equation 3 by the global loss function $\Phi(\cdot)$.

**Assumption 2.4.** *(Constrained PL Type Inequality) For initialization $\underline{\mathbf{w}}^0$ and radius $\rho > 0$, we make the following assumptions on the local and global loss functions:*

1. *The local empirical loss function for each client $k \in [K]$ satisfies (see Theorem D.1)*

$$16\Phi_k(\underline{\mathbf{w}}^0) < \rho^2 \alpha_k(\underline{\mathbf{w}}^0, \rho). \tag{4}$$

2. *The global empirical loss function satisfies (see Theorem 3.1)*

$$\sqrt{128L\Phi(\underline{\mathbf{w}}^0)} < \rho\alpha_g(\underline{\mathbf{w}}^0, \rho), \tag{5}$$

*where $L := \max_{k \in [K]} L_k$.*

**Remark 2.5.** *In the literature, two critical assumptions are commonly made to establish linear convergence of gradient descent (GD): (i) existence of global optimal point $\mathbf{w}^*$ Maralappanavar et al. (2025), Liu et al. (2022); Li et al. (2019), and (ii) strongly convex loss Li et al. (2019); Karimireddy et al. (2020) or loss function satisfying the PL-inequality across entire parameter space (Fan et al., 2023; Maralappanavar et al., 2025). To relax the stringent global PL condition, several works have proposed a local variant, often referred to as the local PL or PL\*-inequality. Under this formulation, the PL inequality is only required to hold within a ball centered around the initialization point. Such relaxations have been explored in non-federated settings(see (Liu et al., 2022; Oymak & Soltanolkotabi, 2019). Despite this relaxation, these works Liu et al. (2022); Oymak & Soltanolkotabi (2019) make a critical assumption on the existence of the global optimal point $\mathbf{w}^*$ in the neighborhood around initialization. In our work, we argue that this assumption can be further relaxed through the novel condition introduced in Assumption 2.4. The new condition for convergence stated above has not appeared in this exact form previously in the FL literature. Importantly, the proposed condition is fundamentally different from both the classical PL and local PL\*-inequalities, as we clarify next.*

**Comparison with PL and Local PL (PL\*) Conditions:** There is a stark difference between the proposed condition Assumption 2.4 and the PL-condition (or PL\* condition), which is defined as $\|\nabla\Phi(\mathbf{w})\|^2 \geq \mu(\Phi(\mathbf{w}) - \Phi(\mathbf{w}^*))$ for some $\mu > 0$ and for all $\mathbf{w} \in \mathbb{R}^d$ (and $\mathbf{w} \in \mathbb{B}[\mathbf{w}^0, \rho]$ for PL\* condition). In the PL condition (and local PL), the constant $\mu$ does not depend on the initialization $\mathbf{w}^0$ and radius $\rho$ as the condition is satisfied universally across the entire parameter space, which is certainly restrictive. Another important assumption made in the local/global PL-condition is the existence of globally optimal solution(s) $\mathbf{w}^*$. In contrast, our proposed condition does not require this assumption; instead, we prove that there exists at least one globally optimal solution under our novel convergence criteria stated in Assumption 2.4. It is important to note that the PL condition must be satisfied over the entire parameter space, which can restrict its applicability to certain loss functions such as logistic loss Karimi et al. (2016). On the other hand, our novel condition is assumed only over a small neighborhood around the initialization, making it more broadly applicable. In addition, restricting the model to a Euclidean ball acts as a natural regularization, which enables us to establish a generalization guarantee.

## 3 Convergence Analysis

It is well known that linear convergence of FedAvg can be achieved if the loss functions satisfy strong convexity or if both local and global loss functions satisfy PL inequality in the entire parameter space (Maralappanavar et al., 2025). In this section, we establish that FedAvg(GD) in Algorithm 1 achieves linear convergence to a global optimum under the novel set of assumptions introduced in Sec. 2. Unlike several prior works (Haddadpour et al., 2019; Stich, 2018), we do not explicitly assume the existence of a global optimal point to establish the linear convergence of FedAvg. In particular, we establish that the conditions stated in Assumption 2.4 not only guarantee linear convergence of Algorithm 1 but also ensure the existence of an optimal solution denoted as $\mathbf{w}^*$ within the closed ball $\mathbb{B}[\underline{\mathbf{w}}^0, \rho]$. The following theorem is a precise statement with proof given in Appendix J.7.

**Theorem 3.1.** *Consider an initialization $\underline{\mathbf{w}}^0 \in \mathbb{R}^{d'}$, radius $\rho > 0$ and the loss function $\Phi : \mathbb{R}^{d'} \to \mathbb{R}$ that satisfy Assumptions 2.2 and 2.4. Then the iterates generated by Algorithm 1 after $R$ communication rounds satisfy*

$$\Phi(\underline{\mathbf{w}}^R) \leq \left(1 - \frac{\eta T \alpha_g(\mathbf{w}^0, \rho)}{4}\right)^R \Phi(\underline{\mathbf{w}}^0), \tag{6}$$

where $\eta = \mathcal{O}\left(\frac{1}{T\alpha_g(\underline{\mathbf{w}}^0, \rho)}\right)$. *Furthermore, there exists* $\mathbf{w}^* \in \mathbb{B}[\underline{\mathbf{w}}^0, \rho]$ *to which the FedAvg(GD) converges, i.e.,* $\lim_{R \to \infty} \Phi(\underline{\mathbf{w}}^R) = \Phi(\mathbf{w}^*) = 0$.

The exact learning rate used in the above theorem satisfies

$$\eta \leq \min\left\{\frac{2}{\alpha_{\min}}, \frac{1}{T\alpha_g}, \frac{1}{L}, \Psi_1, \Psi_2\right\},$$

where $\Psi_1 := \frac{\zeta_\rho \rho}{T\sqrt{2KL\Phi(\underline{\mathbf{w}}^0)}}$ and $\Psi_2 := \frac{\alpha_g \alpha_{\min}}{4TL(4L^2 + L\alpha_{\min})}$ with $L := \max_k L_k$ and $\alpha_{min} := \min_{k \in [K]} \alpha_k$. In the next section, we show that a single hidden layer neural network with random initialization satisfies the conditions of Assumption 2.4 leading to linear convergence of Algorithm 1. The above theorem leads to the following corollary.

**Corollary 3.2.** *By choosing* $\eta$ *as in Theorem 3.1, for any error* $\epsilon > 0$, *Algorithm 1 achieves a loss of* $\Phi(\underline{\mathbf{w}}^R) \leq \epsilon$ *after* $R \geq \mathcal{O}\left(\left\lceil 2\log\left(\frac{\Phi(\underline{\mathbf{w}}^0)}{\epsilon}\right)\right\rceil\right)$ *communication rounds.*

**Note.** Although the above theorem shows linear convergence of Algorithm 1, the classical FedAvg uses SGD updates at the local clients. The above result can be extended to SGD, and a similar result can be obtained, as established in Appendix J. Further, our analysis for the convergence of FedAvg with local SGD steps in AppendixJ does not assume a critical assumption of interpolation[2]. Our next goal is to show the generalization guarantee which involves demonstrating that (a) it is possible to initialize a NN such that it satisfies the convergence criteria provided in Assumption 2.4 so that linear convergence can be obtained and (b) bound the Rademacher complexity of the model so that the generalization error can be made small by increasing $n$ (see Mohri et al. (2019)). Towards this, we consider a single hidden layer NN and prove the generalization guarantee of Algorithm 1. Note that our proof of bounding Rademacher complexity requires the iterates of the algorithm to lie within $\mathbb{B}[\underline{\mathbf{w}}^0, \rho]$. Unfortunately, this cannot be guaranteed if the updates of Algorithm 1 use SGD. Hence, our generalization results assume GD instead of SGD. The extension of our analysis to SGD case is relegated to future work.

## 4 Neural Network (NN) Satisfying Constrained PL Type Inequality

In this section, we show that there exist NNs with a squared loss such that Assumption 2.4 is satisfied, and hence leads to linear convergence of FedAvg(GD) (see Theorem 3.1). Towards this, we consider the following NN with a single hidden layer. We assume that the first layer has $m$ neurons followed by a smooth activation function. The output of this NN is given by Arora et al. (2019)

$$f_{\mathbf{w}}(\mathbf{x}) = \frac{1}{\sqrt{m}} \sum_{j=1}^{m} v_j \sigma(\mathbf{w}_j^\top \mathbf{x}), \tag{7}$$

where $\mathbf{x} \in \mathbb{R}^d$ is the input feature vector. With a slight abuse of notation, we have used $\mathbf{w} = \text{vec}([\mathbf{w}_1, \mathbf{w}_2, \ldots, \mathbf{w}_m]) \in \mathbb{R}^{dm \times 1}$ to denote the aggregated weight vectors in the first layer and $\boldsymbol{v} = (v_1, v_2, \ldots, v_m)^\top$ to denote the weight in the second layer, which are sampled independently and uniformly from $\{-1, 1\}$. In addition, we make the following assumption about the activation function, which is used to upper-bound the output of the neural network equation 7.

**Assumption 4.1.** *The activation* $\sigma : \mathbb{R} \to \mathbb{R}$ *is a smooth non-decreasing function with* $\sigma(0) = 0$ *such that the* $1^{st}$- *and* $2^{nd}$ *order derivatives of* $\sigma$ *are bounded, i.e.,* $|\sigma'(x)| \leq D_\sigma$ *and* $|\sigma''(x)| \leq \Delta_\sigma$.

Note that the above condition is satisfied by the tanh activation function, i.e., $\sigma(x) = \tanh(x)$. With $\sigma(x) \neq 0$, many activation functions such as Softmax (see Xu et al. (2015)) satisfy the conditions mentioned in Assumption 4.1. Condition $\sigma(0) = 0$ is assumed for simplicity and ease of calculations. It is worth noting that the well-known ReLU activation does not satisfy the smoothness condition, but it can be well approximated by a smooth proxy function (see (Xu et al., 2015)). Towards establishing the Assumption 2.4, we make the following assumption on the dataset $\mathcal{Z}_k$ at each client.

---

[2]Interpolation refers to the existence of a $\mathbf{w}^*$ such that $\Phi_{k,i}(\mathbf{w}^*) = 0$ for all $k \in [K]$ and $i \in [n]$.

**Assumption 4.2.** *The samples at client $k \in [K]$ denoted by $\mathcal{Z}_k = \{(\mathbf{x}_{k,1}, y_{k,1}), \ldots, (\mathbf{x}_{k,n}, y_{k,n})\}$ are assumed to satisfy $y_{k,i} \leq y_{max}$ and $||\mathbf{x}_{k,i}||^2 \leq 1$ for all $k \in [K]$ and $i \in [n]$*[3].

### 4.1 Analysis of Single Hidden Layer NN

In order to establish Assumption 2.4, we consider the global empirical loss defined in equation 2 for single hidden layer NN equation 7, where $\Phi_k : \mathbb{R}^{md} \to \mathbb{R}$ is the squared loss for each client $k \in [K]$ defined as

$$\Phi_k(\mathbf{w}) = \sum_{i=1}^{n} [f_{\mathbf{w}}(\mathbf{x}_{k,i}) - y_{k,i}]^2 = \|\boldsymbol{e}_k\|_2^2,$$

where $\boldsymbol{e}_k \in \mathbb{R}^n$ and the $i^{\text{th}}$ entry of the error vector $[\boldsymbol{e}_k]_i = [f_{\mathbf{w}}(\mathbf{x}_{k,i}) - y_{k,i}]$. By using notation $\boldsymbol{e} = [\boldsymbol{e}_1^\top, \boldsymbol{e}_2^\top, \ldots, \boldsymbol{e}_n^\top]^\top \in \mathbb{R}^{nK}$, the global loss can be written in compact form as

$$\Phi(\mathbf{w}) = \frac{1}{K} \sum_{k=1}^{K} \|\boldsymbol{e}_k\|^2 = \frac{\|\boldsymbol{e}\|^2}{K}. \tag{8}$$

The gradient of the loss function $\Phi_k$ with respect to the model weight $\mathbf{w} = \mathsf{vec}([\mathbf{w}_1, \mathbf{w}_2, \ldots, \mathbf{w}_m]) \in \mathbb{R}^{dm \times 1}$ is given by: $\nabla_{\mathbf{w}} \Phi_k(\mathbf{w}) = \boldsymbol{J}_k(\mathbf{w})^\top [f_{\mathbf{w}}(\mathbf{x}) - y]$. The Jacobian matrix is defined as

$$\boldsymbol{J}_k(\mathbf{w}) = \frac{1}{\sqrt{m}} \times \boldsymbol{H}_k(\mathbf{w}), \tag{9}$$

where each entry of $\boldsymbol{J}_k(\mathbf{w})$ is a $d$-dimensional row vector, and $\boldsymbol{H}_k(\mathbf{w})$ is defined as

$$\boldsymbol{H}_k(\mathbf{w}) := \begin{bmatrix} v_1 \sigma'(\mathbf{w}_1^\top \mathbf{x}_{k,1}) \mathbf{x}_{k,1}^\top & \cdots & v_m \sigma'(\mathbf{w}_m^\top \mathbf{x}_{k,1}) \mathbf{x}_{k,1}^\top \\ \cdot & \cdot & \cdot \\ \cdot & \cdot & \cdot \\ \cdot & \cdot & \cdot \\ v_1 \sigma'(\mathbf{w}_1^\top \mathbf{x}_{k,n}) \mathbf{x}_{k,n}^\top & \cdots & v_m \sigma'(\mathbf{w}_m^\top \mathbf{x}_{k,n}) \mathbf{x}_{k,n}^\top \end{bmatrix} \in \mathbb{R}^{n \times md}. \tag{10}$$

Similarly, we define the Jacobian matrix $\boldsymbol{J}(\mathbf{w})$ for the global loss function equation 2 by stacking the Jacobian matrices of local loss functions, $\boldsymbol{H}_k^\top(\mathbf{w})$, row-wise as

$$\boldsymbol{J}(\mathbf{w}) = \frac{1}{\sqrt{m}} \times \boldsymbol{H}(\mathbf{w}), \tag{11}$$

where $\boldsymbol{H}(\mathbf{w}) := \left[ \boldsymbol{H}_1^\top(\mathbf{w}), \boldsymbol{H}_2^\top(\mathbf{w}), \ldots, \boldsymbol{H}_K^\top(\mathbf{w}) \right] \in \mathbb{R}^{md \times Kn}$. Next, we discuss the conditions under which the local loss for the single hidden layer NN satisfies Assumption 2.4. To formalize these conditions, we introduce the following quantity, defined in terms of the Jacobian evaluated at initialization $\underline{\mathbf{w}}^0 \in \mathbb{R}^{md}$:

$$\lambda_{k,\rho}^-(m) := \inf_{\mathbf{w} \in \mathbb{B}[\underline{\mathbf{w}}^0, \rho]} \frac{\boldsymbol{e}_k^\top \boldsymbol{H}_k(\underline{\mathbf{w}}^0) \boldsymbol{H}_k(\underline{\mathbf{w}}^0)^\top \boldsymbol{e}_k}{\|\boldsymbol{e}_k\|^2} \tag{12}$$

for each client $k \in [K]$. We also define the corresponding global quantity aggregated over all clients:

$$\lambda_\rho^-(m) := \inf_{\mathbf{w} \in \mathbb{B}[\underline{\mathbf{w}}^0, \rho]} \frac{\boldsymbol{e}^\top \boldsymbol{H}(\underline{\mathbf{w}}^0)^\top \boldsymbol{H}(\underline{\mathbf{w}}^0) \boldsymbol{e}}{\|\boldsymbol{e}\|^2}. \tag{13}$$

Here, the error vectors $\boldsymbol{e}_k$ and $\boldsymbol{e}$ depend on the model parameter $\mathbf{w} \in \mathbb{B}[\underline{\mathbf{w}}^0, \rho] \subset \mathbb{R}^{d'}$. An exact condition in terms of $\lambda_{k,\rho}^-(m)$ and $\lambda_\rho^-(m)$ for linear convergence is provided in the next section. However, it is important to note that $\lambda_{k,\rho}^-(m)$ and $\lambda_\rho^-(m)$ are positive provided $m \geq n/d$ and $m \geq nK/d$, respectively. Otherwise, the conditions are not met, as the matrices will be rank-deficient. To prove the generalization result, we utilize the following setting.

---

[3]Here, $y_{\max} := \max_{i,k} y_{k,i} \; \forall \; k \in [K]$ and $i \in [n]$.

**Setting 4.1.** The setting consists of a single hidden layer NN with $m \geq nK/d$ and squared error loss. In addition, we use Algorithm 1 with $\underline{\mathbf{w}}^0 \sim \mathcal{N}(\mathbf{0}, \frac{1}{d} I_{md \times md})$.

### 4.2 Condition on the NN

To prove the linear convergence of Algorithm 1 for a single hidden-layer NN, we need to verify the conditions stated in Assumption 2.4. As explained previously, the condition will be stated in terms of the quantities in equation 12 and equation 13. The following theorem provides a condition under which Algorithm 1 converges linearly to a global optimal; the proof can be found in Appendix F. We use the following notations to state the theorem

$$\Psi_{m,K,n,\rho} := \sqrt{bn\left(\frac{8\lambda_\rho^+(m)}{m} + \frac{d\Delta_\sigma^2\rho^2}{m}\right)} \quad \text{and} \quad b := \frac{2D_\sigma^2\rho^2 d\log(2n/\delta)}{m} + 2y_{\max}^2,$$

with $\lambda_\rho^+(m) := \sup_{\mathbf{w} \in \mathbb{B}[\underline{\mathbf{w}}^0, \rho]} \frac{\|\boldsymbol{H}(\mathbf{w}^0)\boldsymbol{e}\|^2}{\|\boldsymbol{e}\|^2} \leq \lambda_{\max}\left(H_k(\underline{\mathbf{w}}_0)H_k(\underline{\mathbf{w}}_0^\top)\right)$. It turns out that the maximum eigenvalue scales linearly with $m$ and $n$ (see Appendix G).

**Theorem 4.3.** *Under Setting 4.1 and Assumption 2.2, the loss functions satisfy Assumption 2.4 with a probability of at least $1 - \delta/2$, for any $\delta > 0$ provided the following holds:*

$$\frac{\lambda_{k,\rho}^-(m)}{m} > 2 \times \left[\frac{\Delta_\sigma^2 d\rho^2}{m} + \frac{4bn}{\rho^2}\right], \tag{14}$$

*and*

$$\frac{\lambda_\rho^-(m)}{m} > \frac{4K\Psi_{m,K,n,\rho}}{(1-\zeta_\rho)\rho} + \frac{2d\Delta_\sigma^2\rho}{m}, \tag{15}$$

*where $\lambda_{k,\rho}^-(m)$ and $\lambda_\rho^-(m)$ are as defined in equation 12 and equation 13, respectively.*

To the best of our knowledge, these conditions are the first of their kind. Given the conditions of the theorem, we can ensure that Algorithm 1 converges linearly for a single hidden-layer NN. However, to ensure that the conditions in Theorem 4.3 are satisfied, we first need to verify if the right-hand side expressions in equation 14 and equation 15 can be made small (by choosing the appropriate $\rho$ and $m$) while keeping the left-hand side positive for all $m$. A theoretical observation similar to the assumption below was made in (Telgarsky, 2021, Page 39). The experimental justification is provided in Sec. 6 and additional experiments in Appendix A.2.

**Assumption 4.4.** *We assume that both $\lambda_{k,\rho}^-(m)$ and $\lambda_\rho^-(m)$ scale as $\mathcal{O}\left(\frac{m}{n}\right)$.*

Given the above assumption, we can ensure that the ratios $\lambda_{k,\rho}^-(m)/m$ and $\lambda_\rho^-(m)/m$ are constants independent of $m$ but decrease with $n$. Further, note that the terms $\lambda_{k,\rho}^-(m)/m$ and $\lambda_\rho^-(m)/m$ are less sensitive to $\rho$ since they are upper bounded by the smallest eigenvalue of $\boldsymbol{H}_k(\mathbf{w}^0)^\top\boldsymbol{H}_k(\mathbf{w}^0)$ and $\boldsymbol{H}(\mathbf{w}^0)^\top\boldsymbol{H}(\mathbf{w}^0)$, respectively. In particular, these eigenvalues depend on the initialization $\underline{\mathbf{w}}^0$ while the original condition is in terms of the ball around the initialization. Hence, using the eigenvalues in place of $\lambda_{k,\rho}^-(m)$ and $\lambda_\rho^-(m)$ in the new conditions makes verification straightforward. The higher values of $\rho$ make the right-hand side in equation 14 large, and therefore conditions may not be satisfied, as expected. On the other hand, the same can also be observed for smaller values of $\rho$. The following corollary establishes that the conditions are satisfied by choosing appropriate values of $\rho$ and $m$. The precise parameter settings ensuring these conditions are provided in Appendix F.1.

**Corollary 4.5.** *Choosing $\rho = c \times \mathcal{O}(n^2)$ and $m = V \times \mathcal{O}(\rho^2 n^2)$ in Theorem 4.3 ensures that the conditions in equation 14 and equation 15 are satisfied for sufficiently large $V$ and $c$.*

The above corollary shows that by choosing a large radius $\rho$, a large number of nodes in the second layer, and a sufficiently large $n$, linear convergence can be guaranteed. This presents several challenges in proving the generalization guarantee, especially in proving a bound on the Rademacher complexity.

## 5   Generalization Performance of Single Hidden Layer NN

In this section, we show that Algorithm 1 under **Setting** 4.1 employing a single hidden-layer NN exhibits impressive generalization guarantees. To state the generalization result, we need the notion of Rademacher complexity of the single hidden layer NN (see Mohri et al. (2018)). For an FL setting, the generalization guarantee is provided in Mohri et al. (2019), and the result requires the loss to be bounded. In our case, the loss is bounded (due to the smoothness assumption and the iterates lying within a ball) but can potentially be large, and grow with the radius $\rho$, which is undesirable. We handle this by focusing on the ensemble of "good" NNs, $\mathcal{G}_{\boldsymbol{v}} \coloneqq \left\{ \boldsymbol{v} \in \{-1, 1\}^m : |\sum_{i=1}^n \zeta_i f_{\mathbf{w}}(\mathbf{x})| < \Delta, \quad \forall \quad \mathbf{w} \in \mathbb{B}[\underline{\mathbf{w}}^0, \rho] \right\}$, i.e., $\boldsymbol{v} \in \mathcal{G}_{\boldsymbol{v}}$, whose output scales as $\mathcal{O}(d)$. Note that the class of NNs is in terms of the final layer parameters $\boldsymbol{v}$. Here, $\Delta \coloneqq \sqrt{2} D_\sigma d \sqrt{\frac{\rho^2 + 4m}{m} \log 4}$ and $\mathbf{x}$ is any data point sampled from client-specific data distribution $\mathcal{D}_k(\mathbf{x})$. Focusing on the ensemble of "good" NNs helps us to bound the Rademacher complexity and show that it behaves as $\mathcal{O}(1/\sqrt{n})$ (see Appendix C.1). In Appendix C.1, using the fact that the weight vector lies within a ball of radius $\rho$ around $\underline{\mathbf{w}}^0$, we show that there exist such NNs with output bounded by $\Delta$. For the above reasons, we redefine the Rademacher complexity in terms of conditional expectation.

**Definition 5.1** (**See Mohri et al. (2019)**). *The constrained Rademacher complexity of a class of single hidden layer NN with weights $\mathbf{w} \in \mathbb{B}[\underline{\mathbf{w}}^0, \rho]$ at client $k \in [K]$ is defined as*

$$Rad_k(\underline{\mathbf{w}}^0, \rho) \coloneqq \mathbb{E}_c \left[ \sup_{\mathbf{w} \in \mathbb{B}[\underline{\mathbf{w}}^0, \rho]} \frac{1}{n} \sum_{i=1}^n \zeta_i f_{\mathbf{w}}(\mathbf{x}_{k,i}) \right],$$

*where the conditional expectation $\mathbb{E}_c$ is with respect to $\boldsymbol{\zeta} \coloneqq (\zeta_1, \zeta_2, \ldots, \zeta_n)$ sampled independently and uniformly from $\{-1, +1\}^n$ conditioned on $\boldsymbol{v} \coloneqq (v_1, v_2, \ldots, v_m)^T \in \mathcal{G}_{\boldsymbol{v}}$.*

We use the above definition to derive the following generalization theorem (Mohri et al. (2019)), proof of which can be found in Appendix H. Denote the population (true) loss as $\mathcal{L}(\mathbf{w}; \boldsymbol{v})$ and the empirical loss as $\mathcal{L}_{\mathcal{Z}}(\mathbf{w}; \boldsymbol{v}) \coloneqq \Phi(\mathbf{w})/n$, where $\mathcal{Z} = \mathcal{Z}_1, \ldots, \mathcal{Z}_K$ is the combined dataset with samples drawn i.i.d. from $\mathcal{D}_k$. Further, let $\mathcal{L}(\mathbf{w}, \boldsymbol{v}) \coloneqq \mathbb{E}\mathcal{L}_{\mathcal{Z}}(\mathbf{w}; \boldsymbol{v})$, where the expectation is with respect to the data. Define

$$\Psi \coloneqq \left( \frac{2(\rho^2 + 4m) D_\sigma^2 d^2}{m} + y_{max}^2 \right) \sqrt{2 \log \left( \frac{1}{\delta} \right)},$$

where $y_{max}$ and $D_\sigma$ are as defined in Assumptions 4.2 and 4.1, respectively.

**Theorem 5.2** (Generalization Bound). *For the setting in 4.1, the single hidden layer NN achieves the following generalization bound with a probability of at least $1 - \delta$:*

$$\mathcal{L}(\mathbf{w}; \boldsymbol{v}) \leq \mathcal{L}_{\mathcal{Z}}(\mathbf{w}; \boldsymbol{v}) + \frac{2}{K} \sum_{k=1}^K Rad_k(\underline{\mathbf{w}}^0, \rho) + \frac{\Psi}{\sqrt{nK}}.$$

We next provide a tractable upper bound on the constrained Rademacher complexity for the class of single-hidden-layer neural networks satisfying the proposed conditions.

**Theorem 5.3.** *The constrained Rademacher complexity of client $k \in [K]$ is bounded by*

$$Rad_k(\underline{\mathbf{w}}^0, \rho) \leq \frac{1}{n\sqrt{m}} + \sqrt{\frac{\nu D_\sigma^2 d^2 \log \left( \frac{N_{\theta, \rho}}{\delta_1} \right)}{n}},$$

*where $\nu = (\rho^2 + 3m)/m$, $N_{\theta, \rho} \coloneqq 3d^{3/4}\sqrt{\rho D_\sigma nm}$ and $\delta_1 \coloneqq \frac{1}{2mn\sqrt{2}D_\sigma d}\sqrt{\frac{m}{(\rho^2 + 4m)}}$.*

*Proof.* Refer to Appendix I.                                                                                                   □

It is clear from Theorem 5.2 that a good generalization can be achieved if (a) the empirical loss $\mathcal{L}_{\mathcal{Z}}(\mathbf{w};\boldsymbol{v})$, (b) the Rademacher complexity, and (c) $\frac{\Psi}{\sqrt{nK}}$ are all small.

**Small empirical loss:** Note that the empirical loss $\mathcal{L}_{\mathcal{Z}}(\mathbf{w};\boldsymbol{v})$ depends on the conditions stated in Theorem 4.3 and the communication rounds. The conditions in Theorem 4.3 can be ensured by choosing $\rho = c\mathcal{O}(n^2)$ and $m = V \times \mathcal{O}(\rho^2 n^2)$, as shown in Corollary 4.5. Now, by choosing the communication rounds $R \geq \mathcal{O}\left(\left\lceil 2\log\left(\frac{\Phi(\mathbf{w}^0)}{\epsilon}\right)\right\rceil\right)$ as in Corollary 3.2 ensures a small empirical loss, i.e., $\mathcal{L}_{\mathcal{Z}}(\mathbf{w};\boldsymbol{v}) < \epsilon, \forall \epsilon > 0$.

**Small Rademacher complexity and $\frac{\Psi}{\sqrt{nK}}$:** From Theorem 5.3, the first term of the Rademacher complexity bound can be made small by choosing large $n$ and $m = \mathcal{O}(n^6)$ (see Corollary 4.5).[4] Noting that $\delta_1 = \mathcal{O}(1/mn)$ and $\log\left(\frac{N_{\theta,\rho}}{\delta_1}\right) = \mathcal{O}(\log(mn))$, we can ensure that the Rademacher complexity decreases as $\mathcal{O}(1/\sqrt{n})$. The term $\frac{\Psi}{\sqrt{nK}}$ can be made small by choosing large $n$ and recognizing that $\Psi = \mathcal{O}(1)$.

The above discussion results in the following corollary, which, to the best of our knowledge, is the first result of its kind for an FL setting.

**Corollary 5.4.** *With a probability of at least $1 - \delta$, there exists a single hidden layer NN employing the FedAvg(GD) algorithm with sufficiently large $m$, $n$, and $R$ that achieves a generalization error that scales as $\mathcal{O}(1/\sqrt{n})$.*

**Effect of Number of Clients** $(K)$**:** From the generalization bound in Theorem 5.2, it is evident that the last term decreases with $K$ as $1/\sqrt{K}$. However, for larger values of $K$, the learning rate is impacted by $K$ through $\frac{\zeta_\rho \rho}{T\sqrt{2KL\Phi(\mathbf{w}^0)}}$, which scales as $1/\sqrt{K}$ (see Theorem 3.1). From equation 6, the loss goes down as $\exp\{-\mathcal{O}(R/\sqrt{K})\}$ leading to slower convergence. Thus, the overall effect of increasing $K$ on the generalization is insignificant, which is demonstrated in our experimental results in Sec. 6.

# 6 Experimental Evaluation

In this section, we corroborate our theoretical results with empirical evidence. We evaluate our federated learning framework on three widely used datasets, MNIST, Fashion-MNIST, and CIFAR-10. *We have added additional experimental results in Appendix A.*

## 6.1 Empirical Verification of Theorem 4.3

To examine the requirement of Theorem 4.3—that equations 14 and 15 are satisfied, we design the following experimental study. We consider the MNIST, Fashion-MNIST, and CIFAR-10, distributing each dataset across five clients under both homogeneous and heterogeneous partitions. To keep the eigenvalue computations tractable on a 16 GB GPU, we cap each client at 300 samples. Our model is a single-hidden-layer network with a tanh activation, and we vary the width of the hidden layer from 1000 to 2600 to place the network firmly in the over-parameterized regime. Here, we focus on the client-wise setup; the corresponding federated client setup is presented in the extended experimental results in Appendix A.

**Client-wise: (See Fig. 1)** We randomly select a single client and compute the minimum eigenvalue of the matrix $H_k(\underline{\mathbf{w}}_0)H_k(\underline{\mathbf{w}}_0)^\top$ for varying hidden layer sizes $m$. Note that the minimum eigenvalue exhibits a linear relationship with $m$, indicating that the ratio $\frac{\lambda_{k,\rho}^-(m)}{m}$ remains approximately constant as $m$ increases (see Fig. 1) for both heterogeneous and homogeneous datasets for a fixed value of $n$. Since $\lambda_{\min}\left(H_k(\mathbf{w}_0)H_k(\mathbf{w}_0)^\top\right) \geq \lambda_{k,\rho}^-(m)$, we use the minimum eigenvalue of $H_k(\underline{\mathbf{w}}_0)H_k(\underline{\mathbf{w}}_0)^\top$, i.e., $\lambda_{\min}\left(H_k(\underline{\mathbf{w}}_0)H_k(\underline{\mathbf{w}}_0^\top)\right)$, as a practical surrogate for $\lambda_{k,\rho}^-(m)$ in our empirical analysis. Fig. 1 confirms that $\frac{\lambda_{k,\rho}^-(m)}{m}$ scales at least linearly with $m$ for a fixed $n$; this enables us to choose a large $m$ (i.e., a large $V$) for a fixed $n$ to satisfy the constraint in Theorem 4.3, as demonstrated in Corrollary 4.5.

---

[4]Tightening the dependency of $m$ on $n$ is relegated to future work.

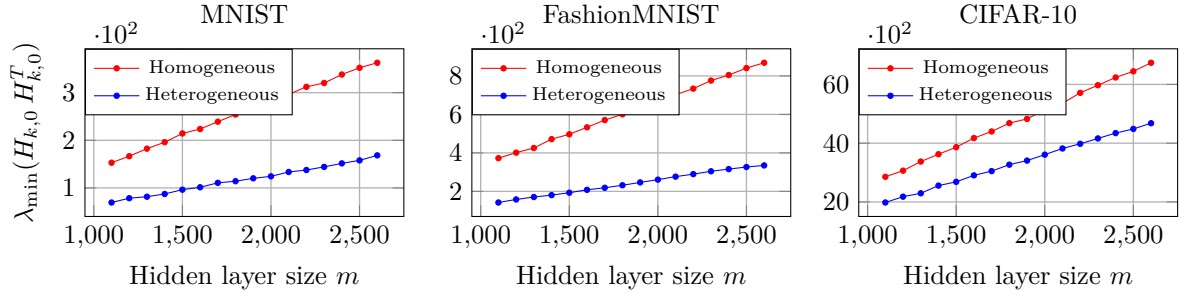

Figure 1: Shows that the ratio $\lambda_{\min}(H_{k,0}H_{k,0}^T)/m$ is constant across clients for MNIST, FashionMNIST, and CIFAR-10 datasets in both homogeneous and heterogeneous settings, where $H_{k,0} = H_k(\underline{\mathbf{w}}_0)$.

## 6.2 Empirical validation of Assumption 4.4

The experiments in Fig. 2a and Fig. 2b demonstrate that $\lambda_{\rho,k}^-(m)$ scales as $\mathcal{O}(m/n)$. We conducted this experiment using the MNIST dataset under two settings: homogeneous and heterogeneous data distributions. An experiment for homogeneous data distribution can be found in Appendix A. A single-hidden-layer neural network with a squared loss function was employed for both cases. For each configuration, we computed the minimum eigenvalue of the matrix $H_{k,0}H_{k,0}^\top$ and visualized its behavior. Since the minimum eigenvalue of this matrix, $\lambda_{\min}(H_{k,0}H_{k,0}^\top)$, is always greater than or equal to $\lambda_{k,\rho}^-(m)$, we use $\lambda_{\min}(H_{k,0}H_{k,0}^\top)$ as a computable surrogate for $\lambda_{k,\rho}^-(m)$ in our empirical evaluation. Figure 1 shows that $\lambda_{\min}(H_{k,0}H_{k,0}^\top)$ grows linearly with $m$, implying that the ratio $\lambda_{\min}(H_{k,0}H_{k,0}^\top)/m$ for a fixed $n$ approaches a constant for large $m$. We then examine its behavior with respect to $n$. As shown in Fig. 2b, for different values of $m$, the quantity $\lambda_{\min}(H_{k,0}H_{k,0}^\top)/m$ decreases with $n$ and eventually saturates to a positive constant. To highlight this trend, we plot $n \times \lambda_{\min}(H_{k,0}H_{k,0}^\top)$ as a function of $n$ for several choices of $m$. We observe that the asymptotic value to which $\lambda_{\min}(H_{k,0}H_{k,0}^\top)$ converges increases with $m$ in Fig. 2b. Together, these observations indicate that the minimum eigenvalue scales proportionally to $n/m$, confirming the $\mathcal{O}(n/m)$ assumed in Assumption 4.4. Equivalently, this means that the quantity $n \times \lambda_{min}(H_{k,0}H_{k,0}^\top)$ saturates to a positive value for sufficiently large $m$ and $n$, as visualized in the surface plot shown in Fig. 2a.

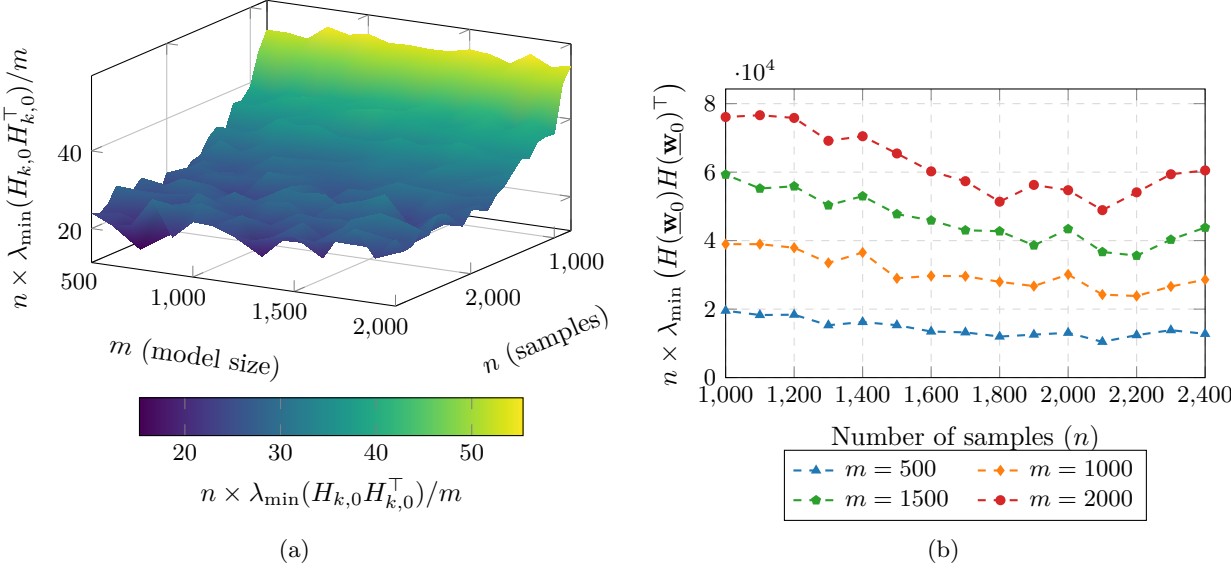

(a)    (b)

Figure 2: **(a).** Surface plot of heterogeneous data showing the relationship between samples $(n)$, model size $(m)$, and scaled minimum eigenvalue. **(b).** Minimum eigenvalue scaled by number of samples $(n \times \lambda_{\min})$ versus number of samples $(n)$ for different values of $m$ with heterogeneous data.

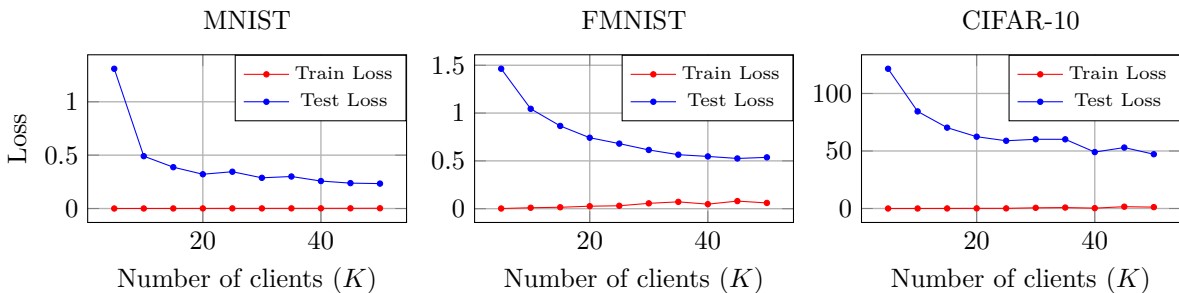

Figure 3: Trend of training and testing error with respect to the number of clients $K$ under heterogeneous data setting for MNIST, FMNIST, and CIFAR-10 datasets.

### 6.3 Effect of Number of Clients

In all our results, we do not make any assumption about the client homogeneity. As a consequence, in Theorem 5.3, we established that the number of clients has an insignificant effect on the generalization error. In this subsection, we provide empirical validation of this result.

**Experimental Setup:** We perform experiments on the MNIST, Fashion-MNIST, and CIFAR-10 datasets under a *heterogeneous data* distribution across clients. The model is a single-hidden-layer neural network with 500 hidden neurons. Each client is assigned 200 data samples so that the network remains in the over-parameterized regime, aligning with the theoretical assumptions.[5]

**Effect of Number of Clients:** We plot the training and test losses as a function of the number of clients $K$. As expected, the training loss remains close to zero in all client configurations under consideration. Interestingly, the test loss initially decreases with $K$ but stabilizes once the number of clients exceeds approximately 20 (see Fig. 3). This observation corroborates our theoretical result (see Theorem 5.2), which establishes that under heterogeneous data distributions, increasing the number of clients beyond a certain threshold has a negligible effect on the generalization error. This is because, beyond that point, the Rademacher complexity dominates the bound. Therefore, to achieve a zero generalization error, one needs to increase the number of samples $n$.

## 7 Conclusion

In this work, we address the problem of generalization, along with convergence guarantees, of the widely used FedAvg algorithm for solving Federated Learning (FL) problems. We proved the generalization bound by handling the optimization error and the Rademacher complexity. The optimization error was handled by proposing a new constrained Polyak-Łojasiewicz (PL) type conditions on the (local) loss functions. Under these new conditions, we showed that there exists a global optimum to which the FedAvg(GD) converges linearly after $\mathcal{O}(\log(1/\epsilon))$ rounds of communication, where $\epsilon$ is the desired optimality gap. Importantly, we demonstrated that a broad class of single-hidden-layer neural networks satisfies the proposed conditions, that are required to establish the linear convergence of FedAvg, as long as $m > \frac{nK}{d}$, where $m$ is the number of neurons in the hidden layer, $n$ is the number of samples at each client, $K$ is the number of clients, and $d$ is the feature dimension. This provides the first theoretical evidence that such networks can yield linear convergence of FedAvg(GD) without assuming global convexity or global PL conditions. Leveraging the fact that all iterates remain confined within a ball centered at the initialization, we bound the Rademacher complexity of this hypothesis class. This allowed us to establish that the generalization error of FedAvg(GD) decreases at the rate $\mathcal{O}(1/\sqrt{n})$. Together, our results provide the first unified framework that simultaneously offers linear convergence and a provably diminishing generalization error for FedAvg(GD) under realistic non-convex settings.

---

[5]The condition requires $m \geq nK/d$ for the eigenvalues to be non-zero.

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
