# OpenReview forum: "Linear Convergence and Generalization of FedAvg Under Constrained PL-Type Assumptions: A Single Hidden Layer Neural Network Analysis"
_TMLR — Rejected by TMLR_

### Review · Reviewer_jT3s · 2026-02-09

**Summary Of Contributions:**

This is a theory paper. It proves that FedAvg converges linearly to the optimal solution if the local loss functions at each client and the global loss function satisfy a constrained PL-type condition. The authors also show that a single hidden-layer NN with squared loss satisfies the conditions proposed in this paper. Additionally, they derive an upper bound on the Rademacher complexity for a class of single hidden-layer NNs and analyze the effect of the number of clients and model size on the generalization performance.

**Audience:**

Yes

**Audience Explanation:**

This is a theory paper. This paper studies both the optimization error and the generalization gap in federated learning, and I believe it would be of interest to the community.

**Claims And Evidence:**

Yes

**Claims Explanation:**

This paper made the following claims:

- FedAvg, when using Gradient Descent (rather than SGD) on the client side and full client participation, converges linearly to the optimal solution if both the local and the global loss functions satisfy the proposed Constrained PL-Type Inequality (Assumption 2.4).
- An over-parameterized single hidden-layer neural network with squared loss can satisfy Assumption 2.4 and can achieve the linear convergence result for FedAvg(GD).
- Using the fact that FedAvg(GD) iterates remain within a rho-ball around the initialization, the authors derive an upper bound on the Rademacher complexity for this NN class.

All of the above claims are supported by theoretical proofs.

Questions:

The authors state that the generalization error, _regardless of data heterogeneity_, diminishes at the rate \bigO(1/\sqrt{n}):
- How exactly is data heterogeneity incorporated into the generalization analysis?
- In particular, how is this handled in the proofs and experiment setup? Even after reading the appendix, it is unclear how heterogeneous settings are formally treated in the experiment section.

Assumption 2.4 is very critical in this work. I am curious in the potential relationship between \alpha_k and \alpha_g. Can one be possibly bounded by another or are they completely independent? Some discussion or intuition could strengthen the understanding.

I am not sure about the meaning of \zeta_\rho in equation 15.

I cannot understand how Figure 1 demonstrates that the ratio is _constant across clients_ for MNIST, Fashion-MNIST, and CIFAR-10. Do the authors mean client across different datasets, or clients within a single federated learning setup?

**Requested Changes:**

See above

---

> ### Author Response · Authors · 2026-02-25
> **Clarification on heterogeneity in generalization error**
>
> We thank the reviewer for the constructive feedback. We would like to clarify the questions raised below, one by one.
> **Question: How exactly is data heterogeneity incorporated into the generalization analysis? In particular, how is this handled in the proofs and experiment setup? Even after reading the appendix, it is unclear how heterogeneous settings are formally treated in the experiment section.**
>
> **Response:** The data heterogeneity is captured implicitly through the local and global regularity constants, denoted by $\alpha_k$ and $\alpha_g$, which appear in our proposed **Constrained PL-type inequality (Assumption 2.4).** These constants govern the curvature structure of the local and aggregated objectives $\Phi_k$ and $\Phi$ respectively, and play a central role in establishing the linear convergence of the optimization error, specifically the first term $\mathcal{L}_Z(\mathbf w; v)$ in the generalization bound of Theorem 4.3.
>
> Traditional federated learning analyses often introduce **explicit heterogeneity measures**, such as bounded gradient dissimilarity,
> $$\|\nabla \Phi_k(\mathbf w) - \nabla \Phi(\mathbf w)\| \le \delta,$$
> for all $k$ to capture the effect of non-IID data across clients. For example, in our analysis **(Section 4.1)**, we have considered an example of single hidden-layer neural network where the local $(\alpha_k)$ and global $(\alpha_g)$ rgularities constants correspond to the quantities $\lambda_{k,\rho}^-$ and $\lambda_{\rho}^-$ defined in Equations $(12)$ and $(13)$ of the main paper.
>
> Specifically, over a neighborhood of the initialization $\mathbf{\underline{w}}^0$, we define
> \begin{equation}
>     \lambda_{k,\rho}^-(m)=\inf_{\mathbf{w}\in \mathbb{B}[\underline{\mathbf{w}}^0,\rho]}\frac{ e_k^\top H_k(\mathbf{\underline{w}}^0) H_k(\mathbf{\underline{w}}^0)^\top e_k}{\| e_k\|^2}
> \end{equation}
> and
> \begin{equation}
>     \lambda_{\rho}^-(m)=\inf_{\mathbf{w}\in \mathbb{B}[\underline{\mathbf{w}}^0,\rho]}\frac{ e^{\top} H(\mathbf{\underline{w}}^0)^\top H(\mathbf{\underline{w}}^0) e}{\| e\|^2},
> \end{equation}
> where $\mathbf e:= [ e_1^{\top}, e_2^{\top}, \ldots, e_n^{\top}]^\top \in\mathbb{R}^{nK}$ and $\mathbf e_k\in \mathbb{R}^n$ with $i$-th entry of $e_k$ is defined by $[e_{k,i}]:=[f_{w}(\mathbf x_{k,i})-y_{k,i}]$. It is important to note that both these terms depend on $\mathbf w$. Note that the regularity constant depends on the local and global Hessian matrices $H_k(\mathbf w^0))$ and $H(\mathbf w^0)$, respectively. When client data are relatively homogeneous, the local Hessians tend to be well aligned, and their corresponding minimum eigenvalues of the gram matrix $H(\mathbf w^0)H(\mathbf w^0)^T$ remain sufficiently large. In this regime, the regularity condition required in Theorem 4.3 is easily satisfied, which ensures the exponential (linear convergence) decrease of the optimization error $\mathcal{L}_Z(\mathbf w;v)$, which is the first term in the generalization bound stated in Theorem 5.2. In contrast, when the data are heterogeneous, the curvature directions across clients can become misaligned. This misalignment leads to smaller minimum eigenvalues of the Hessian-based matrices, effectively weakening the regularity constants. As a result, the sufficient condition for linear convergence becomes harder to satisfy. Thus, rather than quantifying heterogeneity through gradient deviation bounds, our analysis embeds it directly into curvature-dependent regularity parameter $\alpha_g$ that ensures that the optimization error $\mathcal{L}_Z(\mathbf w;v)$ which is the first term of the generalization bound  of Theorem 5.2, goes down at linear rate, while statistical complexity of the hypothesis class is handled separately via Rademacher complexity arguments.
>
> To perform the experiments, we partition dataset accross clients as follows.
> **homogeneous:** Each client has the same number of samples and identical class proportions.
> **heterogeneous:** Each client has the same total samples but draws class proportions from a Dirichlet(alpha) so proportions differ. Refer to data_partition file in submitted code.
> Figure 1 illustrate that when data is heterogeneous, $\lambda_{k,\rho}(H_k(\mathbf w^0)H_k(\mathbf w^0)^T)$ is small whereas for homogeneous data $\lambda_{k,\rho}(H_k(\mathbf w^0)H_k(\mathbf w^0)^T)$ is large and condition is easily satisfied as compared to heterogeneous data.

---

> ### Author Response · Authors · 2026-02-25
> **Clarification on relationship between $\alpha_k$ and $\alpha_g$**
>
> **Question: Assumption 2.4 is very critical in this work. I am curious in the potential relationship between $\alpha_k$ and $\alpha_g$. Can one be possibly bounded by another or are they completely independent? Some discussion or intuition could strengthen the understanding.**
>
> **Response:** It seems that the client-wise regularity constant $\alpha_k$ and the global regularity constant $\alpha_g$ are dependent, however the constant $\alpha_k$ characterizes the local growth of the gradient of the client-wise empirical loss $\Phi_k(\mathbf w)$ within the ball $\mathbb{B}[\mathbf w_0,\rho]$, whereas $\alpha_g$ captures the corresponding property of the averaged global loss $\Phi(\mathbf w) = \frac{1}{K}\sum_{k=1}^K \Phi_k(\mathbf w)$. Although the global gradient is the average of local gradients, i.e.,
> $$\nabla \Phi(w) = \frac{1}{K}\sum_{k=1}^K \nabla \Phi_k(w)$$
>
> this averaging can hide significant disagreement among clients when the data are heterogeneous. If local gradients $\nabla\Phi_k(\mathbf w)$ point in opposing directions, they may cancel each other out, resulting in a small global gradient even though each client experiences a large gradient. In this case, the constant $\alpha_g$ can be substantially smaller than the corresponding client-wise constants $\alpha_k$. In contrast, when client objectives are well aligned, local gradients reinforce each other, and $\alpha_g$ can be comparable to or even exceed individual $\alpha_k$. Therefore, no general bound of the form $\alpha_g \geq c\cdot \alpha_k$ or $\alpha_k \ge c\cdot\alpha_g$ holds without additional assumptions. This lack of a general relationship motivates the formulation of Assumption 2.4, which imposes separate constrained PL-type conditions on both the local and global objectives. These conditions together ensure linear convergence.

---

> ### Author Response · Authors · 2026-02-25
> **Clarification on remaining questions**
>
> **Question: I am not sure about the meaning of \zeta_\rho in equation 15.**
>
> **Response:** You are absolutely right that the definition of $\zeta_\rho$ was not explicitly stated in Theorem 4.3. The constant $\zeta_\rho$ is introduced in Lemma E.4 in the Appendix, where it is shown to be parameter satisfying $0 < \zeta_\rho < 1$. It represents the decay factor derived in Lemma E.4, and is subsequently used in Equation (15). We agree that this should have been clearly stated in the Theorem 4.3 for completeness and readability. In the revised version, we will explicitly define $\zeta_\rho$ in the statement of Theorem 4.3.
>
> **Question: I cannot understand how Figure 1 demonstrates that the ratio is constant across clients for MNIST, Fashion-MNIST, and CIFAR-10. Do the authors mean client across different datasets, or clients within a single federated learning setup?**
>
> **Response:** Thank you for pointing this out, and we apologize for the lack of clarity in the original presentation. In our experimental setup, we consider a federated learning scenario in which each client’s local objective $\Phi_k : \mathbb{R}^d \to \mathbb{R}$
> is defined using a single-hidden-layer neural network. We conduct this analysis separately on three standard datasets: MNIST, Fashion-MNIST, and CIFAR-10.
>
> For each dataset, we construct a federated setup with multiple clients. From this setup, we randomly select one client and compute the Hessian of its corresponding local loss function. We then examine the matrix $H_{k,0} H_{k,0}^\top$ and empirically observe that its minimum eigenvalue scales linearly with the hidden layer width (m), i.e., $\lambda_{\min}\left(H_{k,0}H_{k,0}^\top\right) = c \times m$ for some constant $c>0$. As a consequence, the ratio
>
> $$\frac{\lambda_{\min}\left(H_{k,0}H_{k,0}^\top\right)}{m}$$
>
> remains approximately constant as $m$ varies. Figure 1 illustrates this linear scaling behavior for one randomly selected client within the federated setup, shown separately for MNIST, Fashion-MNIST, and CIFAR-10. Thus, the constancy refers to variation with respect to the network width $m$ for a fixed client, not to compare across different clients.

---

### Review · Reviewer_n2yK · 2026-03-02

**Summary Of Contributions:**

The authors try to provide both optimization and generalization guarantees for FedAvg on non-convex losses, specifically those obtained when training $1$ layer nonlinear NNs. The authors assume that the outer layer weights of the network are fixed, and only train the inner-layer weights.

To address this issue, the authors use a local analysis, where the iterates lie in the ball $B(w_0, \rho)$ with $w_0$ being the initialization and $\rho$ being the radius of the ball.



1. **Optimization Error**: The authors show that if a constrained PL-type condition holds in this ball, i.e., $\frac{||\nabla f(w)||^2}{f}\geq \alpha$ for both the local and global losses, then
    - Theorem 3.1 : A global minimizer achieving $0$ loss exists in the ball.
    - Corollary 3.2 : FedAvg with full-batch updates and mini-batch updates converge to the optimal loss at a rate linear in the number of rounds.
2. **Validity of the constrained PL-type assumption (Theorem 4.3)**: For $ \frac{nK}{d}\leq m \leq O(n^4)$, the authors show that the above constrained PL type assumption holds for $1$ layer NNs. Here, $K$ is the number of machines, $n$ is the number of local datapoints, $d$ is the dimension of features and $m$ is the width of the $1$ layer network. The assumptions required for this are --
    - Lipschitz and Smooth Activation (Assumption 4.1).
    - Bounded features and labels ($x,y$) with square loss for regression (Assumption 4.2).
    - The minimum non-zero eigen value of the Hessian of the network output is $O(\frac{m}{n})$, both locally and globally(Assumption 4.4).
    - Gaussian initialization for inner-layer weights (Setting 4.1).
3. **Generalization Error**:
    - Theorem 5.2: The authors show that if the output of the NN is bouned, the generalization error is bounded by  Rademacher complexity of the network restricted to the ball averaged over the clients, along with a term that decreases with $nK$, the total number of datapoints.
    - Theorem 5.3 : The authors show that the averaged Rademacher complexity over the ball is bounded by a term that decreases with $n$, the number of local samples.
4. **Experimental Evidence**: The authors train $1$-layer NN on homogeneous and heterogeneous client splits of MNIST, FashionMNIST and CIFAR10 datasets. They verify that $1$-layer NN satisfies the constrained PL type conditions (Figure 1), it's Hessian of outputs scale as $O(\frac{m}{n})$ (Figure 2), and that after increasing the number of clients beyond a certain threshold, the test error is independent of total number of samples and only depends on number of local samples (Figure 3).



## Strengths --
The strengths of the paper are --
1. **Optimization**:
    - Showing global minimizer exists achieving $0$ loss under the constrained PL type assumptions.
    - Establishing convergence of FedAvg under these conditions.
2. **Experimental Evidence**: Verifying that NN actually hold these constrained PL type assumption.
3. **Presentation**: The paper was well-written.

## Weaknesses --
The core weaknesses of this work are **not discussing the connection between constrained PL type assumption  and the existing NTK PL assumptions for 1-layer NNs** in the centralized settings, and **vacuous generalization bounds**.

1. **Connection to NTK assumptions:** The large-width regime that the authors consider, the local PL like assumptions that the authors use, the scale of their outer-layer weights $\frac{1}{\sqrt{m}}$, bounding the hessian's minimum non-zero singular values for $1$-layer NNs,  along with the linear convergence of GD under these assumptions has already been established in (Arora et al 2019, Du et al 2019). Such analyses have been commonly used for NTK (Neural Tanent Kernel) analysis of NNs, initially started by (Jacot et al 2018). The authors do not appropriately discuss this connection (restricted to the paragraph before Section 3). This makes it seem like this is a novel assumption and analysis. In my opinion, the only novelty here lies in showing that such an assumption is sufficient without explicitly assuming existence of a $0$ loss point, and it's extension to the federated settings. There are also certain works for FL in NTK regime which the authors have not discussed (see (Jian & Liu 2025) and the references therein.)

2. **Vacuous Generalization Bounds in overparametrized settings**: The authors make several errors in their analysis of generalization bounds. Fixing these errors makes the generalization bounds vacuous. In addition, the generalization bounds considered in their paper can never be non-vacuous for their settings.
    - In the overparametrized regime that the authors consider, there could be several models in side the ball that achieve $0$ loss, but not all of them generalize well. The generalization bounds based on Rademacher complexity, like the ones in (Mohri et al 2019), are uniform-convergence bounds. (Nagarajan et al 2019) showed that any uniform convergence bound will be vacuous in the overparametrized settings of training NNs.
    - The exact error in the authors' proof is on page 36, the size of the cover. The authors set it to $N_{\theta, \rho} = (\frac{3\rho \sqrt{d}}{\theta})^d$. This is incorrect. The ball $B(w_0, \rho)$ lies in $R^{md}$ as there are $m$ weights of $d$ dimensions. Therefore, the size of the cover is proportional to $(1 + \frac{3\rho}{\theta})^{md}$. Due to this additional $m$ term in the coefficient, $\log(N_{\theta, \rho}) = O(md)$, and $m = \Omega(n)$, so their rademacher complexity bounds in Theorem 5.3 are $O(\sqrt{\frac{\log(N_{\theta, \rho})}{n}}) = O(1)$.
    - A common wisdom for overparametrized NNs generalizing well is that inspite of the vacuous uniform convergence bounds, algorithms like GD might provide an "implicit bias" towards solutions that generalize. A long line of work establishes implicit bias in the overparametrized settings for linear and logistic regression, and neural networks. To show that the implicit bias generalizes, which is commonly called benign overiftting, one has to make additional assumptions on the data, like an essentially low-rank structure, see (Bartlett et al 2019) for linear regression and (Misiakiewicz & Montanari 2023) for 1 layer NNs in the NTK regime that you consider.

3. **Other mistakes in the proof:** There are several other errors in the proof. Some of them are quite major, making the results for generalization as well as convergence of FedAvg with mini-batch updates vacuous.
    - **(Major) Theorem 5.2:** The authors derive the generalization bound for any network with bounded output. This bound has a term of the optimization error. The optimization error is small for the iterates of FedAvg, however, the iterates of FedAvg don't necessarily correspond to a network with bounded outputs. Therefore, applying the generalization bound directly for the FedAvg iterates is not valid.
    - **(Major)  Lemma C.2 :** When applying Hoeffding's the authors set $\sum_{i=1}^m (\frac{1}{\sqrt{m}})^2 = \frac{1}{m}$. This is incorrect. If we fix this error, the concentration in terms of $-m\Delta^2$ becomes $\Delta^2$.
    - **(Major) Appendix J:** The authors show that assuming constrained PL-type inequality on just local and global losses implies existence of a $0$ loss point in the ball for FedAvg with mini-batch updates. This is actually not possible. Their proof strategy for the GD case was to show that iterates of FedAvg lie inside the ball and their global loss decreases linearly in rounds, so the limit point of $0$ loss must also lie inside the ball. For mini-batch updates, the authors can show that $E|| w - w_0|| \leq \rho$, where $w$ is any FedAvg iterate, and that the expected loss decreases linearly. For mini-batch FedAvg, the iterates are random variables. If their expected distance to initialization is small, it means that on expectation their values lie inside the ball. For a random sequence, if it's expected value is bounded, then it does not mean that it's value is bounded almost surely. To show existence of limit point inside ball, you need almost sure boundedness.
    - **(Major) Not accounting for $d$:**  The authors implicitly assume that $d=O(1)$ several times in the main draft (Sections 4.2, Section 5) and the appendix (Sections C, F, G and H). This is incorrect, as $d$ can sometimes be as large as $n$. For CIFAR10, $d\approx3K$, which is comparable to $n = 50K$. Further, in the proof, the authors bound $w^\top x\leq d||w||$ (Page 20, Page 32 in Equation 38). In Page 32 Equation 38, they also incorrectly ignore the term of $m$.

**References**-
- (Du et al 2019) Gradient Descent Finds Global Minima of Deep Neural Networks. ICML.
- (Jacot et al 2018) Neural Tangent Kernel: Convergence and Generalization in Neural Networks. NeurIPS.
- (Nagarajan et al 2019) Uniform convergence may be unable to explain generalization in deep learning. NeurIPS.
- (Bartlett et al 2019) Benign Overfitting in Linear Regression. PNAS.
- (Misiakiewicz & Montanari 2023) Six Lectures on Linearized Neural Networks. Arxiv.
- (Jian & Liu 2025) Widening the Network Mitigates the Impact of Data Heterogeneity on FedAvg. ICML.

**Additional Comments:**

None.

**Audience:**

Yes

**Audience Explanation:**

Analysis of FedAvg for real models used in practice, for instance NNs, is necessary to understand the real-world performance of FL.

**Claims And Evidence:**

No

**Claims Explanation:**

Please see above.

**Requested Changes:**

Please address the main weaknesses. Here are some additional comments.

**Generalization bounds:** In my opinion, establishing generalization bounds (basically showing benign overfitting) for FedAvg might require adapting techniques from (Misiakiewicz & Montanari 2023) to the Federated Case, as well as specifying the implicit bias of FedAvg in the NTK regime. I don't think any existing works have completely solved both these problems. The authors will require much more time than the revision to show the above results, however, this is the only way to actually correctly analyze FedAvg in this regime. In case they are not able to do this, I'd recommend removing the mention of the generalization bounds from this paper.

**Presentation**:
 - Please mention that only the inner weights are trained.
 - Theorem 4.3 : $\zeta_\rho$ is not defined.
 - Experiments : What is the exact heterogeneity setting for the experiments? The authors don't specify if it is label heterogeneity or something else.
 - Page 30 before equation (32) : Lemma has not been referenced properly.
 - Appendix J : Can the authors atleast provide the mini-batch update definition for Appendix J. How is the mini-batch chosen? Are the indices sampled randomly with or without replacement?

---

> ### Author Response · Authors · 2026-03-29
> **Connection to NTK assumptions**
>
> We thank the reviewer for the careful reading of our manuscript and for providing detailed and constructive feedback. The comments regarding the connection to the NTK literature, the analysis of the generalization bounds, and the technical issues pointed in the proofs have been very helpful in improving the clarity and rigor of the paper. We address each of the reviewer’s concerns below and will incorporate the suggested clarifications and corrections in the revised version.
> We respond to the reviewer’s comments point-by-point.
> ## Response to "Connection to NTK assumptions"
> The reviewer correctly points out that our analysis is related to the NTK literature (Jacot et al., 2018; Du et al., 2019; Arora et al., 2019). In fact, the spectral properties of the Jacobian used in our analysis are inspired by these works. However, our results differ in several important ways:
>
> **Federated learning setting**
>
> Existing NTK analyses study centralized gradient descent.
> Our work extends these ideas to FedAvg, where local updates are performed on different clients before aggregation.
> The convergence analysis therefore must control both local and global objectives, which introduces new technical challenges.
>
> **No interpolation assumption**
>
> Classical NTK analyses assume the existence of an interpolating solution $w^*$ with zero training loss. In contrast, our constrained PL condition does not assume the existence of such a solution. Instead, we prove that a global minimizer exists inside the ball.
>
> **Local PL condition**
>
> Prior works (Du et al., 2019; Arora et al., 2019) does assume PL inequality. Their result rely on kernel linearization arguments. Wjereas we assume PL inequality to hold in a restricted neighborhood around initialization, which makes the assumption weaker.
> To clarify the connection with NTK literature, we will expand the discussion in Section 1.2 and explicitly compare our assumptions with those used in (Du et al., 2019; Arora et al., 2019).

---

> ### Author Response · Authors · 2026-03-29
> **Generalization bounds**
>
> ## Response to "Generalization bounds"
>
> We thank the reviewer for the careful reading of the generalization analysis and for pointing out the issue in the covering number argument. The reviewer is correct that the parameter space of the neural network lies in $\mathbb{R}^{md}$, and therefore the covering number of the ball $B[w_0,\rho]$ should scale with the dimension $md$ rather than $d$. Incorporating this correction introduces an additional dependence on the network width $m$, which makes the resulting Rademacher complexity bound vacuous in the overparameterized regime considered in our paper. We appreciate the reviewer for identifying this issue.
>
> More broadly, we agree that uniform convergence bounds based purely on hypothesis class complexity can become vacuous in overparameterized neural networks, as discussed in Nagarajan et al. (2019). In light of this observation, we have revisited the generalization analysis.
>
> In particular, we believe that a more appropriate approach in this setting is to analyze generalization through **algorithmic stability**, which focuses on the behavior of the learning algorithm rather than the size of the hypothesis class. Given that the iterates of FedAvg remain confined to a closed ball $B[w_0,\rho]$, and under our assumptions the network outputs can be controlled, a stability-based analysis appears promising for obtaining meaningful (non-vacuous) generalization guarantees. We are currently working on formalizing this approach.
>
> However, incorporating a complete and rigorous stability-based analysis would require substantial changes to the current manuscript and a significant restructuring of Section 5.
>
> Therefore, depending on the reviewer’s preference, we are happy to proceed in either of the following ways in the revised version:
>
> 1. **Remove the current generalization section entirely** and focus the paper on the optimization analysis and constrained PL framework, or
> 2. **Replace the current generalization analysis with a stability-based framework**, once fully developed and rigorously established.
>
> We would be glad to follow whichever direction the reviewer and the action editor consider more appropriate for the scope of this work.

---

> ### Author Response · Authors · 2026-03-29
> **Other mistakes in the proof:**
>
> 1. We thank the reviewer for pointing out the need to explicitly justify the bounded output assumption used in Theorem 5.2. Our generalization analysis considers neural networks whose parameters lie within the ball $B[w_0,\rho]$. From Theorem 3.1, the iterates of FedAvg remain confined to this ball throughout training. Under Assumptions 4.1 and 4.2, the activation function is Lipschitz and the input features are bounded. These properties imply that the network output $f_w(x)$ is bounded for all $w \in B[w_0,\rho]$.
> In the revised version, we will add a lemma explicitly showing that the boundedness of the parameter vector together with the Lipschitz activation function guarantees a bounded network output. This ensures that the conditions required for Theorem 5.2 are satisfied by the iterates produced by FedAvg.
>
> 2. We appologies for the confusion regarding the summation. While using Hoeffding's inequality, we encounter with following summation:
> $$\sum_{l=1}^m[\frac{2\|\mathbf w_l\|dD_\sigma}{\sqrt m}]^2$$. This is same as
> $$\frac{4d^2D_\sigma^2}{m}\sum_{l=1}^m\|\mathbf w_l\|^2$$.
> By definition of Euclidean norm, this will imply,
> $$\frac{4d^2D_\sigma^2}{m}\|\mathbf w\|^2.$$
> Then we hve added and subtracted $\underline{\mathbf w}^0$ inside norm and used triangle's inequality and so on. Nowhere, we have used $$\sum_{i=1}^m(1/\sqrt m)^2=1/m.$$
>
> 3. We thank the reviewer for carefully analyzing the argument in Appendix J and for pointing out this issue.
>
> We agree with the reviewer that, in the mini-batch (stochastic) setting, the iterates of FedAvg are random variables, and bounding their expected distance from the initialization does not imply almost sure boundedness. Therefore, the current argument is insufficient to guarantee that the iterates remain inside the ball (B[w_0,\rho]) almost surely, and consequently does not justify the existence of a limit point within the ball.
>
> In the revised version, we will correct this issue by restricting our theoretical guarantees to the deterministic FedAvg (full-batch) setting, where the iterates are deterministic and can be shown to remain inside the ball. The extension to mini-batch updates will be clearly stated as future work, which would require a more refined high-probability or almost-sure analysis.

---

> > ### Comment · Reviewer_n2yK · 2026-04-08
> >
> > For the choice of either removing the generalization bounds or keeping the ones on stability, I'm ok with whatever the authors decide as long as it is correct and offers some non-trivial insight.
> >
> > A note about stability based bounds, I'm not sure you can get non-trivial non-vacuous bounds using stability for NNs in the overparametrized regime without early stopping or decreasing step sizes. For non-convex problems, even in the centralized case, the best possible bound on function values grows with the number of iterations for a decreasing step size. Also, I'm not sure if the authors have sufficient time before the final decision to complete this analysis.

---

> > > ### Comment · Reviewer_n2yK · 2026-04-08
> > >
> > > I'd recommend uploading a revision with the requested changes that have been promised.

---

> > > > ### Author Response · Authors · 2026-04-11
> > > > **Re-submission**
> > > >
> > > > We thank the reviewer for the careful reading and constructive feedback.
> > > >
> > > > We acknowledge that completing a rigorous and non-vacuous stability analysis within the final decision timeline is challenging.
> > > >
> > > > Therefore, if the Area Chair and the other reviewers agree, we are ready to submit a revised version that excludes the generalization guarantees and focuses entirely on the optimization aspects of the problem.

---

### Review · Reviewer_4ZU9 · 2026-03-15

**Summary Of Contributions:**

This work studies the convergence rate and generalization of the FedAvg algorithm.
For the convergence rate,
- The authors showed that if the loss functions satisfies a local-PL-type property round the initial point, then a unique global minimizer exists and FedAvg converges linearly to that global minimizer.
- The authors also demonstrated that a single-hidden-layer neural network (NN) satisfies the proposed property if the width is large enough and an additional assumption (Assumption 4.4) is fulfilled.

For generalization,
- The authors proved that the generalization error of the single-hidden-layer NN trained by FedAvg decays at an $O(n^{-1/2})$ rate if the sample size $n$ and the width are large enough (also under Assumption 4.4).

**Audience:**

Yes

**Audience Explanation:**

Yes.
Although this is not my primary area of expertise, FL is an important field of research within both the optimization and machine learning communities.
I find the impact of the number of clients on generalization ability particularly interesting.

**Broader Impact Concerns:**

There are no ethical concerns.

**Claims And Evidence:**

No

**Claims Explanation:**

In my opinion, the results of the paper are valuable;
however, I believe several statements are inaccurate and should be revised.
Some connections with existing works are also missing.

**Comparison with work [1]**
- Although this paper is cited in the introduction, it is not discussed or compared with the current work.
In particular, [1] also analyzes the generalization error without assuming strong convexity and boundedness.
A comparison with [1] should be included in the introduction.

**Connections with works [2,3]**
- In Section 1.1 (Our Contributions), the authors claim novelty of the constrained PL-type condition introduced in this paper.
However, if I'm not mistaken, this property appears to be the same as the local-PL-type property introduced in [2], simply applied to the federated learning (FL) setting. Furthermore, [2] demonstrates the linear convergence of gradient descent and the existence of global minimizers near the initial points.
- In my view, Theorem 3.1 could be seen as a (possibly nontrivial) extension of the result in [2] to FL.
Consequently, I personally do not think the introduction of Assumption 2.4 constitutes a novel contribution, and the connections with [2] should be explained in more detail.
For example, is the analysis more involved than that of [2]?
Does the analysis require specific insights unique to FL?
I will appreciate any clarifications from the authors.


**Assumption 4.4**
- In Section 1.1 (Our Contributions), it is claimed that a single hidden-layer NN with squared loss satisfies the two conditions proposed in Assumption 2.4 (the local-PL-type inequality).
This is not entirely precise, as subsequent theoretical results rely on an additional assumption (Assumption 4.4) that is not justified theoretically.
- Additionally, is Assumption 4.4 used in the statement of Corollary 5.4? Its proof is missing from the Appendix. If it is used, please include the assumption in the statement and provide the proof in the appendix.

**A claim about Rademacher complexity**
- Section 1.1 mentions that "the Rademacher complexity approaches zero if the raidus $\rho=O(\sqrt{n})$ and $m=O(n^3)$.""
I do not see how this follows from Theorem 5.3.


**Claims in numerical experiments**
- In Section. 6.2, it is mentioned that the quantity $\lambda_{\min}(H_{k,0}H_{k,0}^\top)/m$ stablizes at a positive constant as the sample size $n$ increases in Figure 2(b).
However, the results in Figure 2(b) are not that convincing, particularly when $m=1500,2000$.
- In Section 6.3, it is cliamed that the test loss stabilizes once the number of clients exceeds 20.
I find the results in Figure 3 are not very convincing in supporting this claim.

**References**
1. Huang et al. Understanding Convergence and Generalization in Federated Learning through Feature Learning Theory. ICLR 2024.
2. Chatterjee. Convergence of gradient descent for deep neural networks. 2022.
3. An and Liu. Convergence of stochastic gradient descent under a local Lojasiewicz condition for deep neural networks. J. Mach. Learn. 2023.

**Requested Changes:**

**Comparison with work [1].**
Please add a discussion regarding the results of [1].

**Connections with works [2,3].**
Please explain the connection between the current work with [2,3] in more detail.
I would appreciate further clarification regarding the novelty of the constrained PL-type inequality (Assumption 2.4).


**Assumption 4.4.**
Please revise the claim in Section 1.1 and the statement of Corollary 5.4.
Additionally, please include the proof of Corollary 5.4 in the appendix.


**A claim about Rademacher complexity.**
I would appreciate any explanation of the claim that "the Rademacher complexity approaches zero if the raidus $\rho=O(\sqrt{n})$ and $m=O(n^3)$."


**Comments on typos and typesetting.**
Here are some comments regarding the typesetting:
1. In Algorithm 1 (FedAvg), please include $w_k^{r,0}=\underline{w}^r$ for completeness.
2. In Assumption 2.4, it is unclear why "(see Theorem D.1)" is included in the statement.
Please provide an explanation or remove it to avoid confusions.
3. Section 3 mentions that the proof of Theorem 3.1 is provided in Appendix J.7, but it is actually in Appendix E. Please correct this.
4. The constant $\zeta_\rho$ is not defined in Theorem 3.1 (or I was unable to find the definition).
5. Please remove the *Proof* environment after Theorem 5.3 for consistency.
6. In the first line of Section 6.3, it should be $\lambda_{k,\rho}^-$ rather than $\lambda_{\rho,k}^-$.
7. Throughout the paper, the citation styles are inconsistent.
Many citations lack parentheses.
When a citation is not part of the sentence structure, please use \citep (in natbib) or \parencite (in biblatex).

**Comments on Appendix E.1 (Proof of Theorem 3.1).**

1. Section E.1 mentions that the sequence $\{\underline{w}^r \}$ is a Cauchy sequence, but I could not find this statement within the proofs.
If my understanding is correct, this property is crucial to ensure the existence of the global minimizer in Theorem 3.1.
Please clarify this point.
2. In Lemma E.3, E.4 and E.5, the assumptions regarding $\eta$ are unclear.
For instance, the proof of Lemma E.3 relies on Lemma E.2.
Therefore, Lemma E.3 should specify that $\eta<\frac{\alpha_g\alpha_{\min}}{4TL^2(4L+\alpha_{\min})}$.
Please include all necessary assumptions for $\eta$ in the statements of Lemma E.3, E.4 and E.5.
3. In the statement of Lemma E.3, should the exponent on the right hand side of the inequality be $r$ instead of $R$?
4. In Section E.3 (proof of Theorem 3.1), how does the last inequality follows from Assumption 2.4?

---

> ### Author Response · Authors · 2026-03-29
> **Comparison with work [1]**
>
> We thank the reviewer for pointing out the need for a clearer comparison with Huang et al. (2024). We agree that this is an important related work that should be discussed in more detail.
>
> Huang et al. (2024) study the generalization of federated learning through the lens of feature learning theory, focusing on how learned representations evolve and impact generalization performance without relying on strong convexity assumptions. In contrast, our work takes a different approach based on optimization geometry. Specifically, we introduce a constrained PL-type condition to analyze the convergence behavior of FedAvg and establish linear convergence guarantees under this condition.
>
> While both works aim to understand generalization in federated learning without strong convexity, the methodologies and focus differ significantly: Huang et al. analyze feature dynamics and representation learning, whereas our work focuses on optimization properties and the conditions under which FedAvg exhibits linear convergence.
>
> In the revised version, we will expand the introduction and related work sections to include a detailed comparison with Huang et al. (2024), clarifying these differences and positioning our contribution more precisely.

---

> ### Author Response · Authors · 2026-03-29
> **Connections with works [2,3]**
>
> We thank the reviewer for pointing out the connection to the local PL (Łojasiewicz-type) conditions studied in Chatterjee (2022) and An & Liu (2023). We agree that our constrained PL-type condition is closely related in spirit to these existing notions, and we will revise the paper to clarify this connection more explicitly.
>
> Our goal is not to claim that the underlying inequality is entirely new in isolation, but rather to extend such conditions to the federated learning setting and analyze their implications for FedAvg. In particular, the key differences from [2,3] are as follows:
>
> 1. **Federated setting:** Unlike [2,3], which consider centralized gradient descent, our analysis involves multiple local objectives $\Phi_k$ and their aggregation. This requires controlling both local and global loss functions simultaneously, which introduces additional technical challenges due to client-wise updates and aggregation.
>
> 2. **Coupled local and global conditions:** Assumption 2.4 imposes conditions on both the local losses $\Phi_k$ and the global loss $\Phi$. The interaction between these conditions is essential to establish linear convergence of FedAvg, and does not arise in the centralized setting.
>
> 3. **Convergence within a restricted region:** Our analysis explicitly ensures that the iterates remain within a ball $\mathbb B[w_0,\rho]$ and establishes the existence of a global minimizer within this region, without assuming its existence a priori.
>
> We agree that Theorem 3.1 can be viewed as a nontrivial extension of results such as [2] to the federated setting, and we will revise the introduction to reflect this more accurately and avoid overstating novelty.

---

> ### Author Response · Authors · 2026-03-29
> **Assumption 4.4**
>
> We thank the reviewer for pointing out the imprecision regarding Assumption 4.4 and the claims in Section 1.1.
>
> We agree that the statement in Section 1.1 that a single hidden-layer neural network satisfies the conditions in Assumption 2.4 is incomplete. The validity of these conditions relies not only on Assumptions 4.1–4.2, but also on Assumption 4.4. In the revised version, we will correct this statement to explicitly include Assumption 4.4 and avoid any ambiguity.
>
> Regarding Assumption 4.4, we note that it captures the scaling behavior of the minimum eigenvalue of the Jacobian-related matrices, which is consistent with observations in the NTK literature and is supported empirically in Section 6.2. We will revise the text to clearly present this assumption as an empirically justified condition rather than a theoretically established result.
>
> We also agree that the role of Assumption 4.4 in Corollary 5.4 is not clearly stated. In the revised version, we will:
>
> * explicitly include Assumption 4.4 in the statement of Corollary 5.4, and
> * provide a complete proof of the corollary in the appendix.
>
> These changes will ensure that all assumptions and dependencies are clearly stated and justified.

---

> ### Author Response · Authors · 2026-03-29
> **A claim about Rademacher complexity**
>
> We thank the reviewer for pointing out this lack of clarity. We agree that the statement in Section 1.1 does not directly follow from Theorem 5.3 as currently presented, and the dependence on (\rho), (m), and (n) is not made explicit.
>
> More importantly, we acknowledge that in the overparameterized regime considered in our work, uniform convergence bounds based on Rademacher complexity may become vacuous, as also pointed out by reviewer n2yK and related literature.
>
> In light of these observations, we will revise the manuscript as follows:
>
> * remove or significantly soften the claim in Section 1.1 regarding the decay of Rademacher complexity, and
> * either remove the generalization section entirely or replace it with an alternative analysis based on uniform stability, depending on the reviewer’s and action editor’s preference.
>
> This will ensure that all claims in the paper are precise and well-supported.
>
> ## Comments on typos and typesetting
> * **Algorithm 1 (FedAvg):** We will include the missing terms for completeness and clarity.
> * **Assumption 2.4:** We agree that the reference to “(see Theorem D.1)” is unclear in its current form. We will either provide a brief explanation of its role or remove it to avoid confusion.
> * **Incorrect appendix reference:** We will correct the reference in Section 3 to point to Appendix E instead of Appendix J.7.
> * **Undefined constant in Theorem 3.1:** We will explicitly define the constant and ensure all quantities are clearly introduced.
> * **Proof environment after Theorem 5.3:** We will remove it for consistency.
> * **Typographical issue in Section 6.3:** We will correct the notation as pointed out.
> * **Citation style:** We will standardize all citations throughout the paper (e.g., using \citep or \parencite where appropriate).

---

> ### Author Response · Authors · 2026-03-29
> **Comments on Appendix E.1 (Proof of Theorem 3.1)**
>
> 1. **Cruciality of Cauchy sequence**
>
> In the proof, we do not need cauchy rather we need to show that the sequence $\underline{\mathbf{w}}^R$ converges to one of the optimal points. We do this as follows:
>
> From the paper, we know that $\underline{\mathbf{w}}^R$ is a bounded sequence and $\Phi:\mathcal{X} \to \mathbb{R}_{\ge 0}$ converges to zero, i.e.,
> $$
> \Phi(\underline{\mathbf{w}}^R) \to 0.
> $$
>
> Define the zero set:
> $$
> \mathcal{W} := \{ \mathbf w : \Phi(\mathbf w) = 0 \} \bigcap \mathbb B[\underline{\mathbf{w}}^0, \rho].
> $$
>
> We show that $\operatorname{dist}(\underline{\mathbf{w}}^R, \mathcal{W}) \to 0$.
>
> ___
>
> Since $\{\underline{\mathbf{w}}^R\}$ is bounded, it has convergent subsequences.
>
> Take any subsequence $\underline{\mathbf{w}}^{R_k} \to \bar{\mathbf w}$.
>
> By continuity of $\Phi$:
> $$
> \Phi(\bar{\mathbf w}) = \lim_{k \to \infty} \Phi(\underline{\mathbf{w}}^{R_k}) = 0.
> $$
>
> Hence, $\bar{\mathbf w} \in \mathcal{W}$, i.e., every limit point is in $\mathcal W$.
> Next, we prove $dist(\underline{\mathbf w}^{R},\mathcal W)\rightarrow 0.$ For that, assume by contradiction, there exist $\epsilon>0$ and a subsequence $\mathbf w^{R_k}$ such that $dist(\underline{\mathbf w}^{R},\mathcal W)\geq \epsilon$.
> Since $\underline{\mathbf w}^{R_k}\rightarrow \bar{\mathbf w}$ and $\bar{\mathbf w}\in\mathcal W$.
> Therefore, $dist(\underline{\mathbf w}^{R_k},\mathcal W)\rightarrow dist(\bar{\mathbf w},\mathcal W)=0$.
> This is contradiction with assumption $dist(\underline{\mathbf w}^{R},\mathcal W)\geq \epsilon$.
> Hence, $dist(\underline{\mathbf w}^{R},\mathcal W)\rightarrow 0$. This implies $\mathcal W$ is non-empty.
>
> 2. We agree that Lemmas E.3–E.5 do not explicitly state all required assumptions (e.g., on $\eta$ and dependencies on earlier lemmas such as Lemma E.2). In the revised version, we will make each lemma self-contained by clearly including all necessary assumptions.
>
> 3. Yes, you are right. The exponent will be $r$ instead of $R$.
> 4. There is a typo in the Assumption 2.4. The correct assumption would be $$\sqrt{128LK\Phi(\underline{\mathbf w}^0)}\leq \rho\alpha_g(1-\zeta_\rho),$$
> where $0<\zeta_\rho<1.$
> Now, the last inequality follows from this.

---

> ### Comment · Reviewer_4ZU9 · 2026-04-11
>
> I appreciate the authors' detailed response.
>
> * Regarding the comparison with works [1, 2, 3] and Assumption 4.4, please revise the manuscript as outlined in the response.
> * Regarding the Rademacher complexity, I am fine with either option.
> * Regarding the proof of Theorem 3.1 (Appendix E), I am satisfied with the proof the authors provided in their response. However, there are a few statements in the manuscript that still need to be revised. In Appendix E.1, the claim that "we have shown that the sequence $\\{ \underline{w}^r \\}_{r=0}^\infty$ is a Cauchy sequence in the closed subset $B[\underline{w}^0,\rho]$ of a complete Euclidean space" is not precise. In Appendix E.3, the limit $R\to\infty$ should not be taken directly, as the entire sequence may not converge. The proof provided in the authors’ response should be used instead.
>
> Overall, I suggest that the authors upload a revised version of the manuscript so that we can ensure the proof is sound.

---

### Decision · Action_Editor_o5dV · 2026-04-15

**Recommendation:** Reject

**Additional Comments:**

As no revised manuscript was submitted after the discussion phase, the reviewers were unable to assess the proposed changes. I therefore recommend rejection of the current version. In light of the constructive rebuttal and the potentially promising core idea, I enabled the option for a future major revision, provided that the major reviewer concerns are fully addressed.

**Audience:**

Yes

**Audience Explanation:**

Yes. The paper studies FedAvg in a concrete non-convex setting and tries to connect optimization guarantees with neural network structure in federated learning. This is a relevant and nontrivial problem, and the reviewers generally agreed that the topic is of interest to the TMLR audience.

**Claims And Evidence:**

No

**Claims Explanation:**

The current submission raises an interesting theoretical question, but the reviewers identified several issues that go beyond minor presentation fixes. In particular, reviewers pointed out substantial problems in parts of the proof arguments and in some of the claimed results, especially in the generalization analysis and in the stochastic or mini-batch extension.

**Resubmission Of Major Revision:**

The authors may consider submitting a major revision at a later time.